# Elevated dust layers inhibit dissipation of heavy anthropogenic surface air pollution

Zhuang Wang[1,2], Cheng Liu[2,3,1,6,7*], Zhouqing Xie[4,3,1,7*] ,Qihou Hu[1*], Meinrat O. Andreae[8,9], Yunsheng Dong[1], Chun Zhao[5], Ting Liu[5], Yizhi Zhu[1,2], Haoran Liu[11], Chengzhi Xing[1], Wei Tan[1,2], Xiangguang Ji[10], Jinan Lin[1,2], Jianguo Liu[1,3]

[1]Key Lab of Environmental Optics & Technology, Anhui Institute of Optics and Fine Mechanics, Chinese Academy of Sciences, Hefei, 230031, China.

[2]Department of Precision Machinery and Precision Instrumentation, University of Science and Technology of China, Hefei, 230026, China.

[3]Center for Excellence in Regional Atmospheric Environment, Institute of Urban Environment, Chinese Academy of Sciences, Xiamen, 361021, China.

[4]Department of Environmental Science and Engineering, University of Science and Technology of China, Hefei, 230026, China.

[5]School of Earth and Space Sciences, University of Science and Technology of China, Hefei, 230026, China.

[6]Key Laboratory of Precision Scientific Instrumentation of Anhui Higher Education Institutes, University of Science and Technology of China, Hefei, 230026, China.

[7]Anhui Province Key Laboratory of Polar Environment and Global Change, University of Science and Technology of China, Hefei, 230026, China.

[8]Max Planck Institute for Chemistry, Mainz, 55128, Germany.

[9]Department of Geology and Geophysics, King Saud University, 11451 Riyadh, Saudi Arabia.

[10]School of Environmental Science and Optoelectronic Technology, University of Science and Technology of China, Hefei, 230026, China.

[11]Institute of Physical Science and Information Technology, Anhui University, Hefei, 230601, China.

*Correspondence to*: Cheng Liu (chliu81@ustc.edu.cn), Zhouqing Xie (zqxie@ustc.edu.cn), Qihou Hu (qhhu@aiofm.ac.cn)

**Abstract.** Persistent winter–time heavy haze incidents caused by anthropogenic aerosols have repeatedly shrouded North China in recent years, while natural dust from west and northwest of China also frequently affects air quality in this region. Through continuous observation by a multi–wavelength Raman lidar, here we found that winter–time aerosols in North China are typically characterized by a pronounced vertical stratification, where scattering non–spherical particles (dust or mixtures of dust and anthropogenic aerosols) dominated above the planetary boundary layer (PBL), and absorbing spherical particles (anthropogenic aerosols) prevailed within the PBL. This stratification is governed by meteorological conditions that strong northwesterly winds usually prevailed in the lower free troposphere, and southerly winds are dominated in the PBL, producing persistent and intense haze pollution. With the increased contribution of elevated dust to the upper aerosols, the proportion of aerosol and trace gas at the surface in the whole column increased. Model results show that, besides directly deteriorating air quality, the key role of the elevated dust is to depress the development of PBL and weaken the turbulence exchange, mostly

by lower–level cooling and upper–level heating, and it is more obvious during dissipation stage, thus inhibiting the dissipation of heavy surface anthropogenic aerosols. The interactions of natural dust and anthropogenic aerosols under the unique topography of North China increases the surface anthropogenic aerosols and precursor gases, which may be one of the reasons why haze pollution in North China is heavier than that in other heavily polluted areas in China.

## 1 Introduction

Booming industrialization and urbanization in China is releasing large amounts of atmospheric anthropogenic pollutants, especially in the Beijing–Tianjing–Heibei (BTH) and surrounding regions, where the air pollution is the highest in the country (Zhang et al., 2019a; Zhang et al., 2019b). Accumulation of air pollutants from stationary and transportation sources and explosive increase of new particles under stagnant weather conditions (Guo et al., 2014; Huang et al., 2014; Zheng et al. 2015) through chemical reaction, such as multiphase chemical formation (Cheng et al. 2016) as well as regional transport (Li et al., 2017), cause $PM_{2.5}$ (particle mass less than 2.5 μm in diameter) mass concentrations to increase several–fold within a few hours. Recent studies have shown that the radiative effect of aerosols reduces solar shortwave radiation, increases the strength of the capping inversion, and enhances the stability of the planetary boundary layer (PBL) (Zhong et al., 2018). Such unfavorable meteorological conditions will enhance the explosive growth of surface air pollutants. Simulation results from atmospheric chemical transport models have also led to similar conclusions (Ding et al., 2016; Huang et al., 2018), i.e., that absorbing aerosols, particularly black carbon (BC), will increase the temperature at the top of PBL and induce a cooling effect near the surface, thereby inhibiting the dispersion of air pollutants.

In addition to BC, dust is also an important source of air pollution. Besides directly acting as an important component of $PM_{10}$ (particle mass less than 10 μm in diameter) and $PM_{2.5}$, it scatters solar shortwave radiation and absorbs longwave radiation and thus leads to a cooling at the earth surface (Xia and Zong, 2009). Compared with the impact of other aerosol types, such as nitrates and sulfates, the effect of dust on decreasing radiation is more serious (Sokolik and Toon, 1996). Recent studies have also shown that dust can function as a reactant or a catalyst affecting atmospheric chemical reactions (Cwiertny et al., 2008). However, the current understanding of the effects of dust on meteorology and air pollution in North China remains insufficient.

To elucidate the role of dust during heavy air pollution, multi–wavelength Raman lidar (RL) was deployed to monitor the vertical structure of atmospheric aerosols with high spatial and temporal resolution. RL can provide several optical parameters of aerosols to distinguish anthropogenic aerosols, dust, and other aerosol types (de Foy et al., 2011; Freudenthaler et al., 2009; Groß et al., 2013; Müller et al., 2007; Tesche et al., 2009), including the aerosol extinction coefficient (EXT), linear volume depolarization ratio (VDR), and lidar ratio (LR). The RL measurements were performed at the Central Weather Bureau Farm (CWBF) since 17 December 2016 (Fig. 1). The CWBF (39.15° N, 115.73° E) is located 120 km southwest of Beijing and approximately 40 km away from the Baoding urban district. It is surrounded by wheat fields, and there are no nearby stationary pollution sources. Combined with Weather Research and Forecasting (WRF) model coupled with Chemistry (WRF–Chem)

simulations and multi–axis differential optical absorption spectroscopy (MAX–DOAS) observations, the mechanism of dust's impact on meteorology and air pollution was explored.

## 2 Measurements and methodology

### 2.1 Raman lidar system

Ground based RL measurements were performed at CWBF during Jan to Mar 2017. The RL was placed in an air–conditioned room to monitor air pollution through the roof skylight in a continuous mode (7 min for data collection with 15–minute intervals). A schematic of the multi–wavelength RL system is shown in Fig. S1 in the supplement. The light source of the RL system uses an Nd:YAG laser (QSmart850) with a pulse repetition rate of 10 Hz, producing two wavelengths: second harmonic generation 532 nm and third harmonic generation 355 nm, with an output energy of 300 mJ and 230 mJ, respectively. The backscatter signals of the Raman, Rayleigh, and Mie scattering were received by a Cassegrain telescope with a diameter of 400 mm and field of view of 0.2 mrad. In addition, the 532 nm return signal was divided into parallel (532p) and vertical (532s) polarization components. Thus, the receiver had 5 channels: 532p, 355 nm Mie scattering channel, nitrogen (387 nm), water vapor (408 nm) Raman scattering channel, and polarization channel 532s. The data collector was a transient recorder (LICEL, TR20–160) with five acquisition channels. For each channel, the signal was acquired in both analog and photon counting modes with a spatial resolution of 7.5 m. Signals from 4000 laser shots were accumulated to produce a single sampled signal profile (approximately 7 min). More details of the RL system can be found in Table 1.

The RL used in this study can provide various aerosol optical parameters, including EXT, VDR, LR, and relative humidity (RH). The VDR distinguishes between non–spherical and spherical particles (Freudenthaler et al., 2009; Tesche et al., 2009), and non–spherical particles are identified by a high VDR (over 15%). The LR is related to the absorption (>70 sr) and scattering (<40 sr) of particles (Müller et al., 2007). The input signal of aerosol optical parameters are provided in Table 1. The relative error was calculated in accordance with the law of error propagation, and primarily depends on the signal–to–noise ratio (Heese et al., 2010) of the input signal given in Table 1. Data with a signal–to–noise ratio of the input signal less than 1 were discarded. Given that the uncertainty of the overlap correction (Wandinger and Ansmann, 2002) was too high below 400 m, data below 400 m were not used for subsequent analysis. The water vapor sounding experiment was conducted on 16 Aug 2017 at the Beijing Observatory near Beijing's South Fifth Ring (39°48′23″ N, 116°28′03″ E). The RH comparison of the RL and radiosonde is provided in Fig. 2, which shows that the RL and radiosonde were consistent in measuring RH at noon and night. The details of data inversion and data validation can be found in Supplementary materials section S1 and our previous studies (Ji et al., 2019).

## 2.2 Multi–axis differential optical absorption spectroscopy

MAX–DOAS was performed at CWBF since Jan 2017. The instruments used for MAX–DOAS include a telescope, two spectrometers [ultraviolet (303–370 nm) and visible (390–550 nm)] with the temperature stabilized at 20 °C, and a computer that acts as a control and data acquisition unit. The elevation angle (1°–6°, 8°, 10°, 15°, 30°, and 90°) of the telescope is controlled by a stepping motor. The scattered sunlight collected by the telescope is redirected by a prism reflector and a quartz filter to the spectrometer for data analysis. MAX–DOAS can retrieve aerosol profiles with the corresponding aerosol properties and trace gas profiles using the measured spectrum information. Further data screening was conducted using the root mean squares of the residuals of the trace gas ($NO_2$) slant column densities. The system is only operational during the day (from 08:00 to 16:00 local time) with a temporal resolution of 15 min and a spatial resolution of 100 m, respectively. The complete description of the MAX–DOAS system and retrieval algorithm can be found in our previous studies (Xing et al., 2017; Xing et al., 2019).

To explore the effects of upper–level dust on low–level anthropogenic aerosols, the percentage of bottom $EXT_{360}$ in total $EXT_{360}$ and percentage of bottom $NO_2$ volume mixing ratio (VMR) in total $NO_2$ VMR measured via MAX–DOAS were used to represent the low–level air pollution. The percentage of bottom $EXT_{360}$ in total $EXT_{360}$ is defined as

$$EXT_{360\_per} = 100\% \cdot \frac{\int_0^{100\,m} EXT_{360}\ (z)\ dz}{\int_0^{800\,m} EXT_{360}\ (z)\ dz} \tag{1}$$

The percentage of bottom $NO_2$ VMR in total $NO_2$ VMR is defined as

$$NO_{2\_per} = 100\% \cdot \frac{\int_0^{100\,m} NO_2\ VMR\ (z)\ dz}{\int_0^{800\,m} NO_2\ VMR\ (z)\ dz} \tag{2}$$

Where z is the height, and the $EXT_{360\_per}$ and $NO_{2\_per}$ are the percentage of bottom $EXT_{360}$ and percentage of bottom $NO_2$ VMR, respectively.

The EXT comparison of RL and MAX–DOAS during our observation period was shown in Fig. 2. The hourly and spatially average EXT from 400 m to 600 m and 600 m to 800 m were selected due to the blind zone of RL and different spatial resolution between RL (7.5 m) and MAX–DOAS (100 m). The comparison of average EXT profile during HPI 1 and HPI 2 between RL and MAX–DOAS was shown in Fig. S2. In general, the EXT comparisons of RL and MAX–DOAS show a reasonably good agreement (R > 0.8), while the slope of linear regression between RL and MAX–DOAS measured EXT is considerably less than 1. Because the sensitivity of the MAX–DOAS measurements decreases with increasing altitude in the troposphere (Frieß et al., 2006). In addition, MAX–DOAS and lidar measurements were made with different geometries (a combination of zenith–sky and off–axis versus zenith–sky only, respectively) and different integration times for completing a set of measurements (15 versus 22 min, respectively), which may also explain part of the differences (Irie et al., 2008).

## 2.3 WRF–Chem simulations

The air pollution and meteorology parameters from 20 Jan to 5 Feb 2017 were simulated by WRF–Chem version 3.6.1. The model domain was centered at 110.68° E, 39.34° N with a 20 km × 20 km grid resolution, encompassing North China, especially the Mongolia region and its surrounding areas. There are 44 vertical layers from the ground level to the top pressure of 50 hPa, in which 17 layers were located below 2 km to well describe the vertical structure of the air pollutants below PBL. The simulation was conducted from 15 Jan to 5 Feb 2017. Each run covered 48 hours and the last 24–hour results were used for the analysis. The initial and boundary conditions of meteorological fields for simulation were adopted from the 6–hour final operational global analysis (FNL) data generated by the National Environmental Prediction Center (NCEP) with a spatial resolution of 1° × 1°. The Multi–resolution Emission Inventory for China (MEIC, http://www.meicmodel.org/, last access: 6 January 2020) (Liu et al., 2015; Li et al., 2014) was used to obtain anthropogenic emissions. The biogenic emissions were calculated online using the Model of Emissions of Gases and Aerosols from Nature (MEGAN) embedded in the WRF–Chem model. The chemical outputs from previous runs were used as the initial conditions for the following run. The first 5 days were simulated and considered as model spin–up period to minimize the influence of the initial conditions. NCEP's ADP global upper air observations (NCAR archive ds351.0 and ds461.0) were assimilated every 6 hours to reproduce the meteorological field more effectively. Details of model configuration options can be found in Table 2 and our previous studies (Liu et al., 2016a).

In addition, to explore the role of dust in aerosol–meteorology interactions and its impact on surface air pollution during the dissipation stage, the simulation period of each heavy pollution incident dissipation stage was performed five simulations with five different initial times. i.e., the simulation period was from 00:00 to 23:00 on 26 Jan 2017, the five initial times were set to 22:00 on 24 Jan 2017, 23:00 on 24 Jan 2017, 00:00 on 25 Jan 2017, 01:00 on 25 Jan 2017, and 02:00 on 25 Jan 2017. Particularly, four–dimensional data assimilation (FDDA) for wind, temperature, and water vapor mixing ratio was not adopted in the five simulations. The average of five simulations of each simulation period was used for the final analysis.

In this study, we selected the MOSAIC aerosol scheme (Zaveri and Peters, 1999; Zaveri et al., 2008), and the analysis variables here were the 3–D mass mixing ratios of the 32 MOSAIC aerosol variables at each grid point. The model includes organic compounds, black carbon, sulfate, nitrate, ammonium, and other air pollutants with four bin size ranges: (1) 3 nm to 156 nm, (2) 156 nm to 625 nm, (3) 625 nm to 2.5 μm, and (4) 2.5 μm to 10 μm. Thus, model simulated $PM_{10}$ concentrations was given as:

$$PM_{10} = \rho_d \sum_{i=1}^{4} (NO_{3i} + SO_{4i} + NH_{4i} + OC_i + BC_i + CL_i + NA_i + OIN_i)$$

(3)

Where i denotes the bin numbers in the MOSAIC aerosol scheme, the $NO_3$, $SO_4$, $NH_4$, OC, BC, CL, NA, and OIN are 3–D mass mixing ratios of nitrate, sulfate, ammonium, organic compounds, black carbon, chloride, sodium, and other inorganic compounds, respectively. The $\rho_d$ is dry air density, which is used to converts the units of 32 MOSAIC aerosol mixing ratios

from µg/kg to µg/m$^3$.

We conducted two parallel experiments using WRF–Chem to investigate the mechanism of the elevated dust layer enhancing the pollution near the ground: 1. without considering the dust (dust_off); 2. with consideration of the dust (dust_on). The dust concentrations is calculated as

$$\text{Dust} \approx \rho_d \sum_{i=1}^{4} (\text{OIN}_{i\text{dust\_on}} - \text{OIN}_{i\text{dust\_off}}) \tag{4}$$

Where the $\text{OIN}_{i\text{dust\_on}}$ and $\text{OIN}_{i\text{dust\_off}}$ represent the other inorganic compounds in each bin when the influence of dust was considered and ignored, respectively. The non–dust particles concentration is defined as

$$\text{Non-dust} = \rho_d \sum_{i=1}^{4} (\text{NO}_{3i} + \text{SO}_{4i} + \text{NH}_{4i} + \text{OC}_i + \text{BC}_i) \tag{5}$$

The concentrations of upper–level suspended dust is calculated as

$$\text{Dust}_{up} = \frac{\sum_{i=l_{pbl}}^{18} \text{Dust}_i \times (z_{i+1} - z_i)}{z_{18} - z_{l_{pbl}}} \tag{6}$$

Where the $\text{Dust}_{up}$ is the suspended dust concentration above the PBL, $l_{pbl}$ is the number of the model layer closest to the PBL, $\text{Dust}_i$ is the dust concentration in each model layer. The height of the 18$^{th}$ model layer is appoximately 2888 m. The turbulence change within the PBL is calculated as

$$\text{Tur\_exch} = \frac{\sum_{i=1}^{l_{pbl}} \text{exch}_i \times (z_{i+1} - z_i)}{z_{l_{pbl}} - z_1} \tag{7}$$

$$\text{Turbulence change} = 100\% \times \frac{\text{Tur\_exch}_{on} - \text{Tur\_exch}_{off}}{\text{Tur\_exch}_{on}} \tag{8}$$

Where the Tur_exch is average turbulent exchange coefficient within the PBL, $\text{exch}_i$ is the turbulent exchange coefficient of each model layer. $\text{Tur\_exch}_{on}$ and $\text{Tur\_exch}_{off}$ are the average turbulent exchange coefficients for the two experiments dust_on and dust_off, respectively.

To ensure the accuracy of the WRF–Chem model, the key meteorology parameters, including temperature, relative humidity, and wind speed/direction were compared with radiosonde data (http://weather.uwyo.edu/, last access: 6 Jan 2020) at Beijing (39.93 ºN, 116.28 ºE, WMO station number 54511). The radiosondes were launched twice a day (08:00 and 20:00 LT) and measured profiles of atmospheric variables such as air temperature, water mixing ratio, wind speed, etc. As shown in Fig. S3, the WRF–Chem model can effectively reproduce the meteorology parameters. Observed hourly surface–layer PM$_{2.5}$ concentrations from 21 Jan to 6 Feb 2017 at Chengde (40.97 ºN, 117.82 ºE, station number 1065A), Zhangjiakou (40.81 ºN, 114.88 ºE, station number 1059A), Beijing (40.14 ºN, 116.72 ºE, station number 1008A), Tianjing (39.03 ºN, 117.71 ºE, station number 1023A), Baoding (38.88 ºN, 115.44 ºE, station number 1055A), Cangzhou (38.30 ºN, 116.89 ºE, station number

1071A), Shijiazhuang (38.14 ºN, 114.50 ºE, station number 1031A), and Hengshui (37.73 ºN, 115.69 ºE, station number 1076A) were compared with the model results from the dust_on case (Fig. S4). The observed $PM_{2.5}$ values were downloaded from the environmental monitoring station (http://beijingair.sinaapp.com/, last access: 5 January 2020). Generally, the WRF–Chem model can reasonably reproduce the evolutional characteristics of the observed $PM_{2.5}$ concentrations in the eight cities (Li et al., 2016; Wang et al., 2019; Gao et al., 2016) (R: 0.52–0.81). Both the observed and simulated $PM_{2.5}$ concentrations exhibit a heavy pollution period from 22 to 26 Jan 2017 and 1 to 5 Feb 2017.

## 2.4 Characteristics of dust, ice clouds and anthropogenic aerosols

Based on RL measurements, the VDR at 532 nm and LR at 355 nm were derived to represent the characteristics of different aerosol types. The VDR can distinguish between non–spherical and spherical particles (Tesche et al., 2009), which is useful to identify ice clouds (Sassen, 1991) and dust layers (Murayama et al., 1999) (the value is typically greater than 20%). Many researchers have reported the VDR of dust and ice clouds (see Table 3). The typical VDR of Asian dust is between 20% and 33%, which can be distributed at different heights. In addition, the different height distributions of Asian dust may be related to the different origins. The dust from Mongolia generally accumulates between 0 and 3,000 m (Sun et al., 2001), and the dust from the Taklimakan Desert is distributed above 5,000 m (Liu et al., 2008; Sun et al., 2001). Unlike dust, ice clouds have a wider VDR distribution between 20% and 60%, and are usually located above 4,000 m. Therefore, distinguishing Asian dust via high VDR is difficult due to the wide height distribution of dust and a VDR comparable to ice clouds (Sakai et al., 2003).

RL can provide independent measurements of backscatter and extinction profiles (Ansmann et al., 1990; Ferrare et al., 1998) to compute LR. As LR is related to the absorption and scattering of particles (Müller et al., 2007; Omar et al., 2009), a higher LR indicates that the particles tend to be more absorbing. The typical value of LR for Asian dust is 40–60 sr (Omar et al., 2009). The LR of Asian dust observed in Beijing is smaller, from 30 sr to 47 sr, and it is usually located below 3,000 m (see Table 3). By contrast, the LR for ice clouds is lower, less than 30 sr. Therefore, a threshold of 30 sr can be set to distinguish between dust and ice clouds (Sakai et al., 2003). The combined VDR and LR can distinguish between dust and ice clouds. Asian dust has a higher VDR (20%–33%) and the LR is usually greater than 30 sr. The VDR of ice clouds is even higher (20%–60%), but the LR is typically less than 30 sr. In addition to Asian dust and ice clouds, the LR and VDR of anthropogenic aerosols also summarized in Table 3. The low VDR of anthropogenic aerosols (less than 10%) indicates spherical particles (Tesche et al., 2009). The high LR of anthropogenic aerosols is very distinct compared with Asian dust and ice clouds, and the values range from 40 sr to 80 sr.

# 3 Results and discussions

## 3.1 Vertical layering of particles in North China

We focused on the transmission, explosive growth, and dissipation of air pollution along with the interactions between aerosol and meteorology in North China. The $EXT_{355}$ (EXT at 355 nm wavelength) measured via RL shows a periodic cycle of 2–5 days, rising rapidly from less than 0.5 km$^{-1}$ in the early stage of each heavy pollution incident (HPI) to 3–5 km$^{-1}$ within 1–2 days (Fig. 3). For the subsequent discussion, the whole observation set was classified into clean stages, cumulative growth stages (CS), and dissipation stages (DS) based on the surface $EXT_{360}$ (EXT at 360 nm wavelength) measured via MAX–DOAS and the surface winds from model simulations. Clean stages are defined as the times when the surface $EXT_{360}$ is less than 0.5 km$^{-1}$. The surface $EXT_{360}$ during CS and DS is typically greater than 0.5 km$^{-1}$, and the surface winds during CS were dominated by southerly weak winds or a static atmosphere, while much stronger northwesterly surface winds were most prevalent in the DS. An entire HPI includes a clean period plus the subsequent CS and DS. During our whole observation (Fig.S5, Fig. S6, Table S1), Nine HPIs has been observed, the VDR in the upper lidar layer of 8 HPIs is significantly higher than that of the lower lidar layer (except HPI 3), indicating the contribution of dust in the upper lidar layer and anthropogenic aerosols in the lower lidar layer, and the aerosols were stratified. The aerosol stratification was most prominent in HPI 1, HPI 2, and HPI 5, HPI 5 lasted for less than 2 days during the whole observation period (Table S1), whereas the other two HPIs persisted for more than 4 days and had peak PM$_{2.5}$ mass concentrations greater than 500 µg m$^{-3}$. Thus, two HPIs (Table 4), namely, 22 to 26 Jan 2017 (HPI 1), and 1 to 5 Feb 2017 (HPI 2) measured via RL and MAX–DOAS, were selected to represent the typical wintertime pollution cycles in North China (Fig. 3, and Fig. 4).

We evaluated the aerosol optical parameters, including the VDR and LR provided by RL during the HPI 1 and HPI 2 in the upper lidar layer (700–1,300 m) and lower lidar layer (400–600 m). Aerosols that accumulated in the upper lidar layer had a relatively broad VDR value (4%–34% in most cases) and LR range of 32–72 sr (34–60 sr in more than 90% of the cases) during HPI 1 and HPI 2 (Fig. 5a and 5b). Therefore, aerosols accumulated in the upper lidar layer are mainly scattering non–spherical particles. We also selected several RH profiles to identify the aerosol types in the upper lidar layer (Fig. 6). All of the available RH values of aerosols in the upper lidar layer was less than 80%, whereas the RH of ice clouds were usually greater than 100% (Ferrare et al., 1998; Sakai et al., 2003). Furthermore, the non–spherical scattering particles in the upper lidar layer during the two HPIs had the same origin (Fig. 1) and also had similar distribution heights (700–1,300 m). Because anthropogenic aerosols also occurred in the upper lidar layer due to the southerly industrial transport, the non–spherical particles in the upper lidar layer during the HPI 1 and HPI 2 are mainly dust or mixtures of dust and anthropogenic aerosols (polluted dust).

By contrast, a low VDR of less than 10% (2%–8% in most cases) in the lower lidar layer was always found during HPI 1 and HPI 2, and a much higher LR (53–85 sr) in the lower lidar layer (Fig. 5c and 5d) indicating the aerosol's trend to be more absorbing (Müller et al., 2007). The RH of pollutants in the lower lidar layer varied from 25% to 85% and increased as the

pollution grew more severe (see Fig. 6). Moreover, aerosols accumulated in the lower lidar layer came from the polluted industrial regions (Zhang et al., 2019a; Zhang et al., 2019b) (Fig. 1). Therefore, these spherical absorbing particles were mainly anthropogenic aerosols. Based on these measured lidar parameters, we conclude that the aerosols in the upper lidar layer primarily consisted of dust or polluted dust, while the aerosols in the lower lidar layer are mainly anthropogenic aerosols.

During the period from 20 Jan to 5 Feb 2017, weak southerly winds (47%) typically prevailed in the lower lidar layer between the polluted periods (Fig. 1), carrying polluted air masses from industrial areas and resulting in a sharp increase in $EXT_{355}$. The strong northwesterly winds in the lower lidar layer from the Gobi desert (37%) and sparsely populated northern mountain areas (16%) were most prevalent in the dissipation stage and clean period, causing $EXT_{355}$ to drop distinctly (Fig. 3d). The average $EXT_{355}$ in the lower lidar layer during the weak southerly wind conditions was 1.76 $km^{-1}$, followed by winds from Gobi desert (1.35 $km^{-1}$) and sparsely populated northern mountain areas (0.62 $km^{-1}$). The measured VDR in the lower lidar layer was relatively low, and fluctuated with the VDR in the upper lidar layer. In the upper lidar layer, strong northwesterly winds (66%) from the Gobi desert prevailed, carrying dust to the CWBF, leading to a significant increment in VDR. The strong northwesterly winds (7%) in the upper lidar layer from the sparsely populated northern mountain areas usually occurred during the period of VDR decline (Fig. 3e). The $EXT_{355}$ in the upper lidar layer is less than 1.5 $km^{-1}$ in the most cases, except that during the period of southerly wind (27%) transmission, $EXT_{355}$ increased considerably. The average $EXT_{355}$ in the upper lidar layer during the weak southerly wind conditions was 1.00 $km^{-1}$, which is clearly higher than that during the winds from Gobi desert (0.66 $km^{-1}$) and sparsely populated northern mountain areas (0.38 $km^{-1}$).

The shift of the origin of the air mass from northerly to southerly, together with a considerable decrease in wind speed, promotes the southerly transport of industrial pollutants and explosive increase of new particles under stagnant weather conditions (Guo et al., 2014; Zheng et al. 2015) through chemical reaction, such as multiphase chemical formation (Cheng et al. 2016), which is conducive to the accumulation of aerosols in the lower and upper lidar layers. The air mass origin in the upper lidar layer shifts from industrial areas to the Gobi desert with a substantially increasing wind speed, driving the increasement of dust concentrations in the upper lidar layer. As a consequence of these shifts, aerosols are stratified in distinct layers, with anthropogenic aerosols in the lower lidar layer and dust or polluted dust in the upper lidar layer. Thus, the meteorological conditions not only regulate the transmission, accumulation, and dissipation of aerosols, but also control the stratification of air pollutants, which is one of the most powerful factors that promote haze pollution in North China.

**3.2 Correlation between elevated dust and surface haze pollution**

Stratified aerosol typically shrouded CWBF during 20 Jan to 5 Feb 2017. The maximum value of VDR in the upper lidar layer usually appeared during the DS. Moreover, the percentage of $EXT_{355}$ of total $EXT_{355}$ in the lower lidar layer during the DS is considerably higher than during the CS and clean periods (Fig. 7). To further investigate the relationship between elevated dust and surface anthropogenic aerosols, HPI 1 and HPI 2 were examined in detail. During HPI 1, the upper dust layer formed slightly later than the accumulation of the anthropogenic aerosols in the lower lidar layer (Fig.3), indicating the formation of

upper dust is independent of the formation of anthropogenic aerosols in the lower lidar layer. At the end of the CS during HPI 1, the air mass in the upper lidar layer was mainly from the northwest, and the wind speed increased significantly. Particularly, the upper VDR and the percentage of bottom $EXT_{355}$ rose considerably. The weak southerly winds in the lower lidar layer rapidly shifted to strong northwesterly winds during DS during HPI 1. The value of VDR reached a maximum, while the percentage of bottom $EXT_{355}$ rose at first, and then declined. The upper dust layer during HPI 2 appeared earlier than the anthropogenic aerosols in the lower lidar layer (Fig. 3). Similar to HPI 1, the northwesterly winds in the lower lidar layer increased significantly during DS in HPI 2, and both upper VDR and the percentage of bottom $EXT_{355}$ reached a maximum.

We selected hourly and spatially (950 m–1,050 m) average VDR as an indicator of dust in the upper lidar layer. Also, the percentage of bottom $EXT_{360}$ in total $EXT_{360}$ and the percentage of bottom $NO_2$ VMR in total $NO_2$ VMR measured via MAX–DOAS were used to represent the air pollution near the ground. We find that the hourly and spatially average VDR roughly correlates with the hourly average percentage of bottom $EXT_{360}$ and percentage of bottom $NO_2$ VMR during HPI 1 and HPI 2 (Fig. 8). This positive correlation suggests that the increase in upper–level VDR is related to the aggravation of the proportion of aerosol and trace gas at the surface in the whole layer.

### 3.3 Mechanism of the elevated dust layer enhances surface air pollution

We conducted two parallel experiments using WRF–Chem to investigate the mechanism by which the elevated dust layer enhances air pollution near the ground, especially during DS: 1. without considering the dust (dust_off); 2. with consideration of the dust (dust_on). In the MOSAIC aerosol scheme, dust is represented by the difference of "other inorganic compounds" (OIN) between dust_on and dust_off, and non–dust particles include nitrate, sulfate, ammonium, organic compounds, and BC.

The dust concentrations is derived from the OIN difference between the two scenarios of dust_on and dust_off (Equation 4), model simulations well reproduced the spatial and temporal variations of dust concentration at CWBF (Fig. 9 and Fig. 10). The PBL height during the CS was usually below 800 m and decreased with the daily accumulation of air pollutants. Dust typically concentrated above the PBL and the fraction of dust in total $PM_{10}$ concentrations increased with height. The lower PBL height led to a reduction of dust entrainment into the PBL from the upper levels, thereby promoting the stratification of aerosol at CWBF. The northwest wind strengthened during the DS, accompanied by a rise of the PBL. The dust concentration within and above the PBL increased significantly, which may be related to the northwesterly transportation and the rise of the PBL. The model simulations show, consistent with the RL observations, that a large amount of suspended dust can be transported from the Mongolia to downstream urban/industrial regions in Northern China, causing a dust layer that covers the anthropogenic aerosols below. Also, the higher VDR (0.3–0.35) value during the DS suggests that dust controls the optical properties of the upper–level aerosol (Freudenthaler et al., 2009). These invisible (at ground level) but common dust aerosols from northwestern China may induce strong aerosol–PBL feedbacks and affect the PBL structure along their transport path (Liu et al., 2002), and may also impede the dissipation of the underlying aerosol.

To explore the role of dust in aerosol–meteorology interactions and its impact on surface air pollution during the DS, we examined the dissipation process during HPI 1 and HPI 2. The suspended dust above the PBL is widely distributed in North China during HPI 1, whereas it is mainly located in the upper air over the BTH region during HPI 2 (Fig. 11a). Surface dust concentrations also increased but clearly less than those within the PBL (Fig. 11b). Unexpectedly, the concentration of surface non–dust particles increased by 0–11.4 $\mu g/m^3$ after the upper–level suspended dust had passed across the downstream urban/industrial regions in Northern China (Fig. 11c). In addition, the gaseous pollutants ($NO_2$) exhibited the same variation (increases by 0–4.4 ppb) as the non–dust particle concentration (Fig. 11d). The relative increment of surface non–dust particle and $NO_2$ concentrations is 0%–21% (Fig. 11e and 11f). This indicates that, in addition to directly acting as an important component of air pollutants, suspended dust can also induce the enhancement of non–dust particles and precursor gases during DS, thus further increasing the surface anthropogenic aerosol concentrations.

The interaction between dust and meteorology appears to be responsible for the enhancement of surface air pollution during DS. The dust layer during DS plays an important role in modifying the temperature vertical structure (Fig. 12a and 12b). The opposing effects of the dust on temperature, a net heating above the PBL and cooling within the PBL, favor formation of a capping inversion and thereby promote aerosol stratification. Consequently, the role of dust in aerosol–meteorology interactions result in more stagnant conditions, with the turbulent exchange coefficient within the PBL falling by over 60%. Similarly, a significant decrease in PBL height was also attributable to the stable stratification (Fig. 12c and 12d). Also, the maximum reduction of surface horizontal winds speed up to 1.2 m/s, the relative attenuation of surface horizontal winds speed is 0%–27% (Fig. S7), indicating that the elevated dust also weakens the surface advection. In addition, there is no active convection activity (Figure S7) during our observed period (Baro et al., 2015; Gao et al., 2013). As a consequence, although the strong northwesterly winds during DS increase the horizontal and vertical diffusion in the atmosphere considerably, the upper–level dust brought in simultaneously by the northwesterly wind strengthens the temperature inversion due to both scattering and absorption of solar radiation, thereby weakening convective motions. Enhanced horizontal and vertical atmospheric stability due to dust during DS hinders the air pollutants from being dispersed and leads to a reduction of the dissipation rates of surface air pollution (Li et al., 2017; Liu et al., 2016b; Wilcox et al., 2016).

The results demonstrate that dust aerosols during DS can substantially affect meteorological conditions by strong radiative feedbacks, and hence increase the surface air pollution (aerosols and precursor gases) by inhibiting the vertical diffusion of air pollutants. Evidently, such a deterioration of surface air quality is ultimately driven by the emission of pollutants, but is also strongly related to the reduced vertical diffusion capacity of the atmosphere. Surface dimming and upper–PBL warming by dust aerosols help strengthen the capping inversion and weaken turbulent mixing (Li et al., 2017). Previous studies have also found that the levels of gaseous pollutants, such as $NO_2$ (Wallace and Kanaroglou, 2009), are closely related to temperature inversion. Changes of atmospheric stability, precursor gases, and solar radiation could significant modify new particle formation (Zhang, 2017) and photochemical reactions (Zhou et al., 2007), which may also contribute to the surface air pollution. The decreasing upper–level dust concentration (usually less than 40 $\mu g/m^3$ in the model) during CS has an insignificant impact on low–level meteorological conditions, while its mixing with anthropogenic aerosols affects upper–level

aerosol optical properties. Moreover, the mixing of dust and anthropogenic aerosols will promote the atmospheric chemical reactions (Cwiertny et al., 2008), and enhance the formation and growth rates of particles (Nie et al., 2014) to strengthen the particle concentrations in the upper lidar layer (Tao et al., 2014), which in turn further enhances the atmospheric stability and promotes the temperature inversion (Reichardt et al., 2002).

## 4 Summary

Our observations clearly show the stratification of aerosols over North China, especially during the DS. Absorbing spherical particles (anthropogenic aerosols) and scattering non–spherical particles (dust or polluted dust) prevailed in the lower and upper lidar layers, respectively. This stratification was primarily determined by the meteorological conditions. Firstly, the air mass origins of the different layers resulted in different aerosol types, where low–level anthropogenic aerosols came from the southerly polluted industrial regions (Wang et al., 2013) and the upper dust layers arrived mostly from Mongolia (Sun et al., 2001). Secondly, unfavorable vertical diffusion conditions, when strong northwesterly winds prevailed above the PBL with southerly air masses within the PBL, produced lengthy and intense temperature inversions and low PBL heights (Tao et al., 2014). The suppressed convection constrained dust into the PBL, which may also have contributed to higher surface relative humidity (Wilcox et al., 2016). These unique and unfavorable meteorological conditions in North China promote the extremely serious haze pollution and lead to a stratification of aerosols. The VDR in the upper lidar layer and the percentage of $EXT_{355}$ in total $EXT_{355}$ in the lower lidar layer during the DS is considerably higher than during the CS and clean periods. Moreover, the increased share of elevated dust to the upper aerosols coincides with the increase in the proportion of surface aerosol and trace gas in the whole layer. Model simulations show that the elevated dust during DS reduces the lower–level atmospheric turbulent mixing and thereby weaken the diffusion and convection of surface aerosols.

During our three–month observations, we captured nine HPIs and eight of them showed meteorological conditions differences in the PBL and in the free troposphere, which has led to the stratification of aerosols. Therefore, aerosol stratification is common in North China. Here we conclude when southerly transmission dominant in the PBL (anthropogenic aerosols) and northwest transportation prevailed in the free troposphere (dust) usually lead to aerosol stratification. Upper dust aerosol induced dust–meteorology interactions, the dust-meteorology interactions mainly includes two aspects. Firstly, the difference in meteorological conditions between the upper and lower lidar layer leads to the aerosol stratification, upper dust and lower anthropogenic aerosols. Secondly, elevated dust alters the atmospheric thermodynamics and stability, mostly by lower–level cooling and upper–level heating, especially during dissipation stage. The suppressed turbulence exchange and decreased in PBL height impede dissipation of persistent heavy haze pollution. The dust-meteorology interactions provide very important information toward a complete understanding of the formation mechanism of winter haze in North China, and may also explain the special multiphase chemistry in this region.

In summary, we use ground–based observations combined with WRF–Chem simulations to investigate the role of dust on meteorology and air pollution in North China, specifically focusing on the dissipation process during persistent heavy air

pollution events over 4 days. Our results show that, elevated dust not only directly affects the air quality, but also worsens the meteorological conditions to impede the rate of dissipation of surface air pollution, which may be one of the reasons why haze pollution in North China is heavier than that in other parts of the country. The interactions between natural dust and heavy anthropogenic surface air pollution events helps us better understand the transmission, explosive growth, and dissipation of persistent winter–time air pollution in North China. In a similar way, considering the extremely strong long–range transport potential of dust aerosol, Saharan dust could affect India (Deepshikha et al., 2006), Europe (Papayannis et al., 2008), and the United States (Prospero, 1999), since Asian dust can even be transported one full circuit around the globe (Uno et al., 2009). Similar stratification and effects should be investigated in other parts of the world that also suffer from severe particulate pollution (Wu et al., 2017).

*Author contributions.* Cheng Liu, Zhouqing Xie, and Qihou Hu conceived and supervised the study; Zhuang Wang analyzed the Raman Lidar data; Yunsheng Dong provided the technical support for Raman Lidar and data inversion recommendations; Zhuang Wang wrote the manuscript with input from Cheng Liu, Zhouqing Xie, and Qihou Hu; Meinrat O. Andreae reviewed and commented on the paper; Chun Zhao, Ting Liu, Yizhi Zhu, and Qihou Hu provided the WRF–Chem model simulations; Haoran Liu, Chengzhi Xing, Wei Tan, Xiangguang Ji, and Jinan Lin provided the MAX–DOAS data; Meinrat O. Andreae, Cheng Liu, Zhouqing Xie, Qihou Hu and Jianguo Liu contributed to discuss the results and revised manuscript.

*Competing interests.* The authors declare that they have no competing interests.

*Code/Data availability.* All code/data needed to evaluate the conclusions in the paper are present in the paper and/or the Supplementary Materials. Additional code/data related to this paper may be requested from the authors.

*Acknowledgments.* The authors acknowledge the National Oceanic and Atmospheric Administration (NOAA) Air Resources Laboratory (ARL) for the provision of the HYSPLIT transport and dispersion model used in this publication. We thank WRF–Chem developers for making the model available to the scientific community. We thank NASA Langley Research Center Atmospheric Sciences Data Center for providing the the CALIPSO data. This research was supported by grants from the National Key Research and Development Program of China (No. 2018YFC0213104, 2017YFC0210002 and 2016YFC0203302), Anhui Science and Technology Major Project (18030801111), National Natural Science Foundation of China (No. 41722501, 51778596, and 41977184), the Strategic Priority Research Program of the Chinese Academy of Sciences (No. XDA23020301), the National Key Project for Causes and Control of Heavy Air Pollution (No. DQGG0102 and DQGG0205), the Major Projects of High Resolution Earth Observation Systems of National Science and Technology (05-Y30B01-9001-19/20-3), Civil Aerospace Technology Advance Research Project, No.Y7K00100KJ.

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

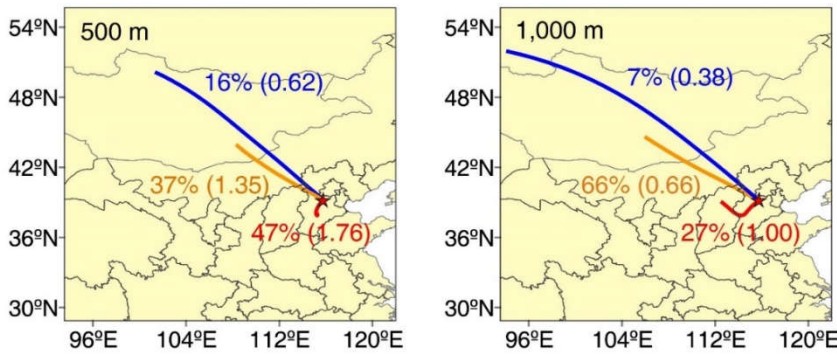

**Figure 1.** Cluster analysis of 24–h air mass backward trajectories (AMBTs) initialized at 500 m and 1000 m from 20 Jan to 5 Feb 2017. The numbers in the map are the fraction of each category of AMBTs and the numbers in brackets are the corresponding spatially average $EXT_{355}$ value at 450 m–550 m and 950 m–1050 m (unit: km$^{-1}$), respectively. The 24–hour AMBTs were computed using the Hybrid Single–Particle Lagrangian Integrated Trajectory (HYSPLIT) model of the National Oceanic and Atmospheric Administration (Draxler and Hess, 1998). We calculated the hourly AMBTs during the whole observation period initialized at 500 m and 1000 m. Then, cluster analysis of AMBTs was conducted in three categories directions. Base map is from TrajStat 1.2.2 software (http://www.meteothinker.com).

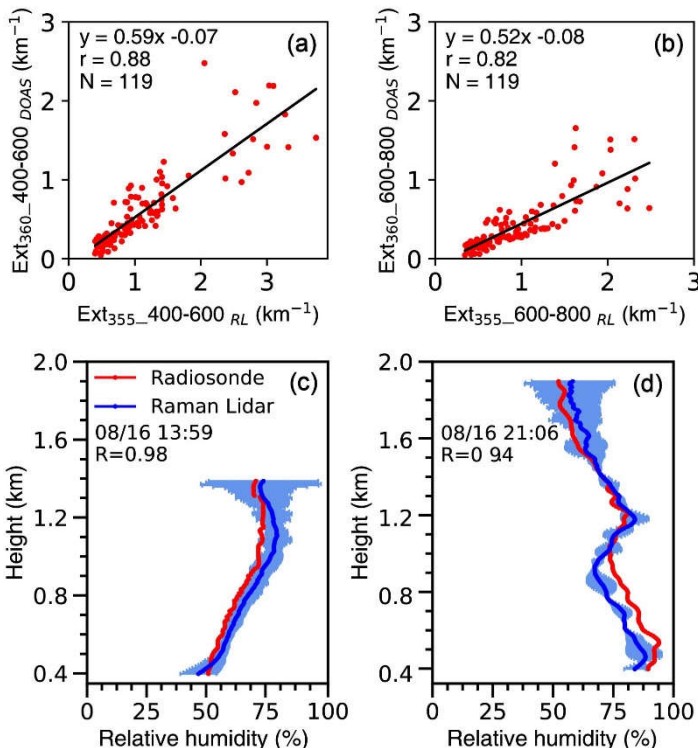

**Figure 2.** Data comparison of RL and MAX–DOAS. Correlations between EXT from MAX–DOAS and RL for layers of (**a**) 400–600 m and (**b**) 600–800 m. RH comparison between radiosonde and RL at (**c**) noon and (**d**) night. The envelopes in (**c**) and (**d**) represent the errors at each altitude. The error is calculated from the law of error propagation, which primarily depends on the signal–to–noise ratio (Heese et al., 2010) of the input signal given in Table 1.

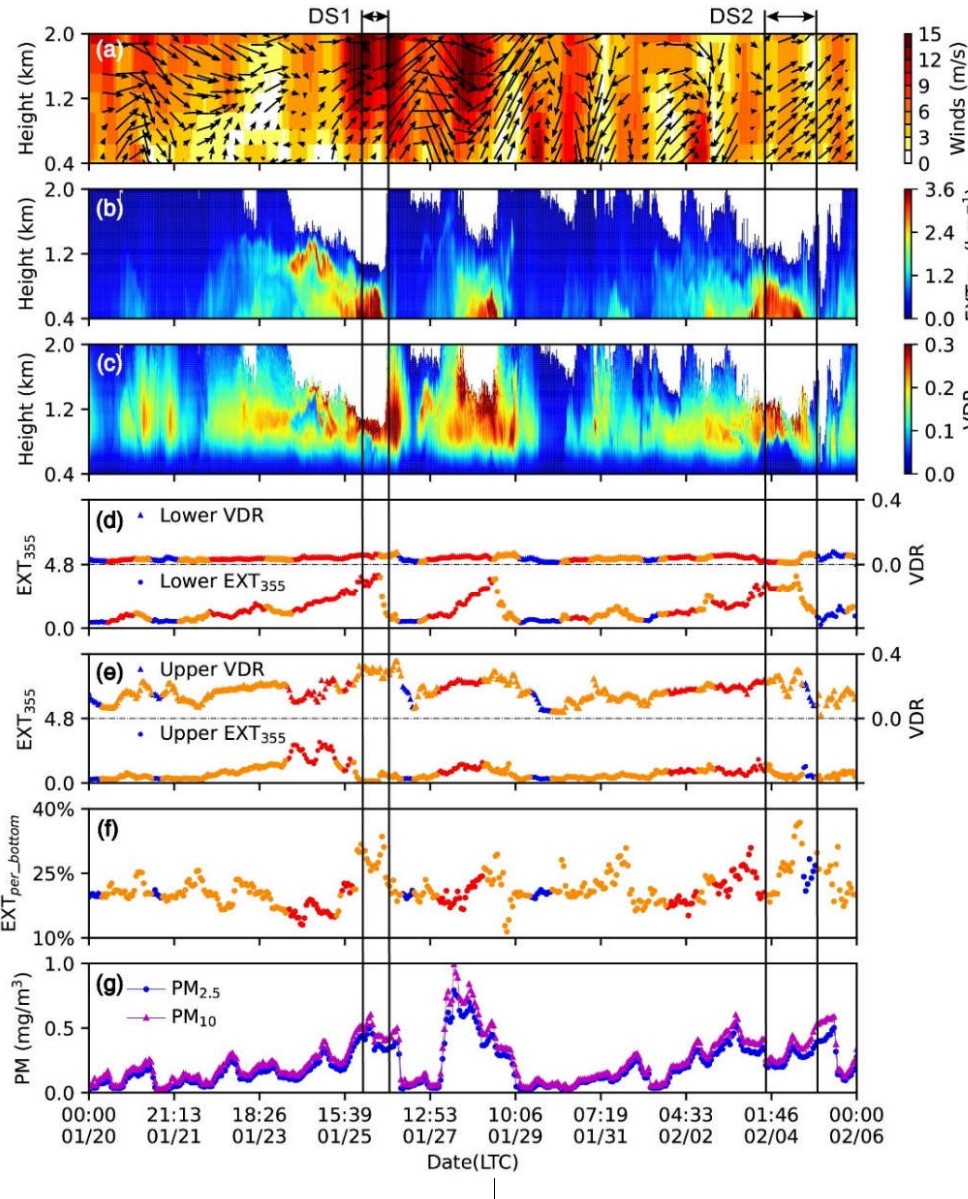

**Figure 3.** Periodic air pollution cycles in North China. The color contours show the vertical structure of (**a**) Horizontal winds simulated by WRF–Chem, (**b**) $EXT_{355}$ and (**c**) VDR measured by RL. Temporal evolutions of spatially average VDR and $EXT_{355}$ at (**d**) 450 m–550 m and (**e**) 950 m–1,050 m. (**f**) The percentage of bottom $EXT_{355}$. (**g**) Temporal evolutions of surface average $PM_{2.5}$ and $PM_{10}$ mass concentrations. The black arrow in (**a**) indicates the wind direction, upper arrow for south winds. The colors in (**c**) and (**d**) represent the air masses originating from southwest (red), Gobi desert (yellow), and sparsely populated northern moutain areas (blue), similar to those in Fig. 1. The observed $PM_{2.5}$ and $PM_{10}$ mass concentrations are the averages of the six environmental monitoring stations in Baoding. The percentage of bottom $EXT_{355}$ is used to characterize the aerosol concentrations in the lower layer, which is defined as: $EXT_{per\_bottom} = 100\% \times \sum\limits_{z=400}^{600} EXT_{355}(z) / \sum\limits_{z=400}^{1000} EXT_{355}(z)$ . Where $EXT_{per\_bottom}$ is the percentage of bottom $EXT_{355}$, z is the height.

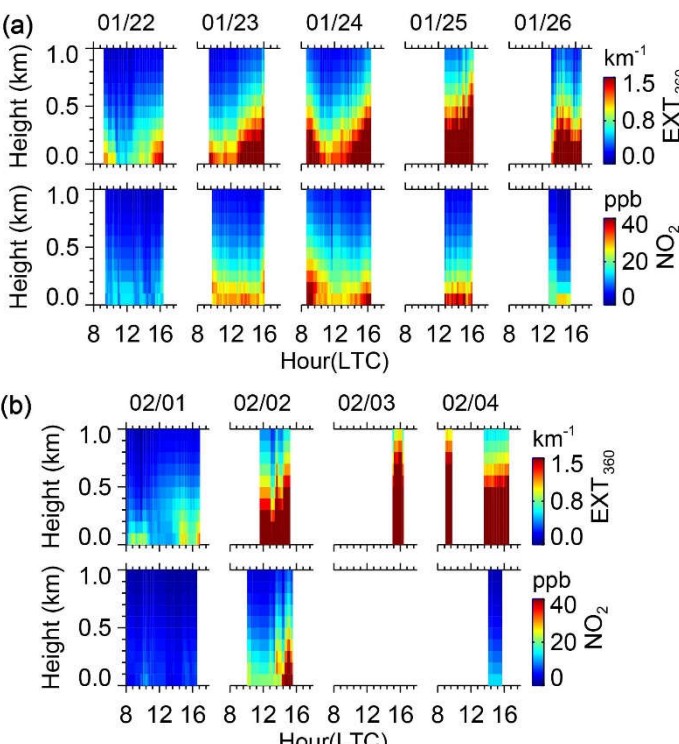

**Figure 4.** Curtain plots of MAX–DOAS observations. (**a**) $EXT_{360}$ and $NO_2$ VMR from 22 to 26 Jan 2017, (**b**) $EXT_{360}$ and $NO_2$ VMR from 1 to 4 Feb 2017.

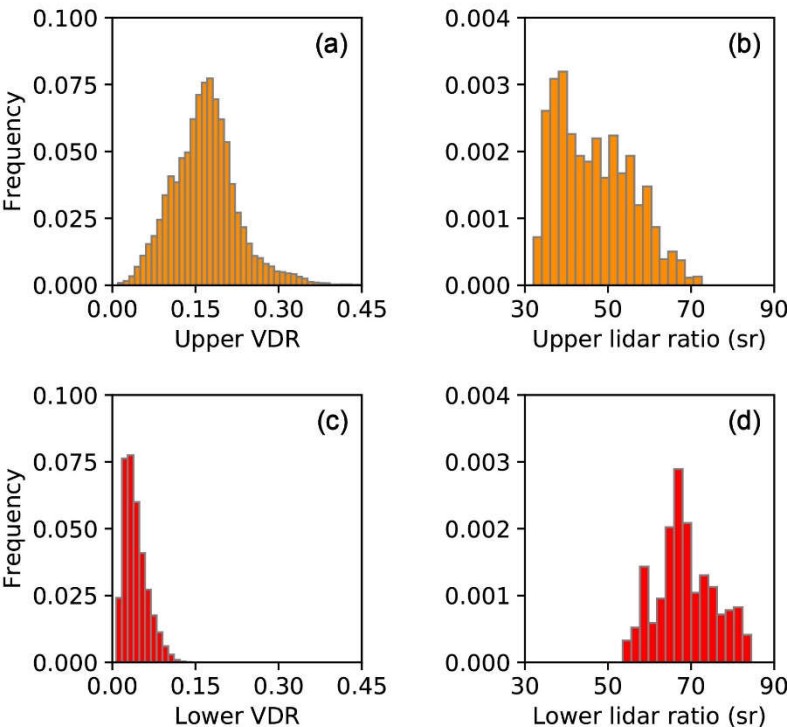

**Figure 5.** Frequency histogram of RL parameters during HPI 1 and HPI 2. Frequency distributions of (**a**) VDR in the upper lidar layer, (**b**) lidar ratio in the upper lidar layer, (**c**) VDR in the lower lidar layer, and (**d**) lidar ratio in the lower lidar layer. The steps of VDR and lidar ratio are 0.01, and 2 sr, respectively.

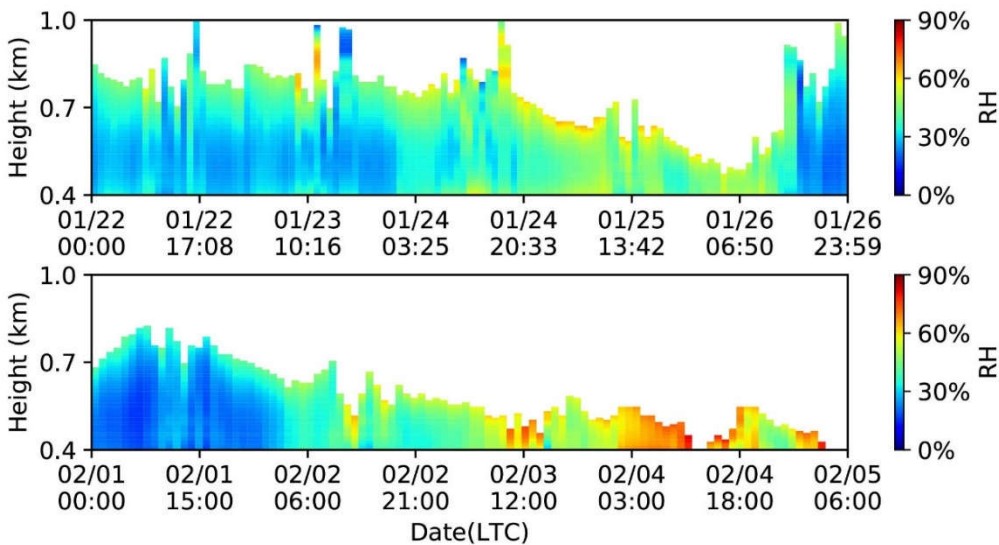

**Figure 6.** Curtain plots of relative humidity during HPI 1 and HPI 2.

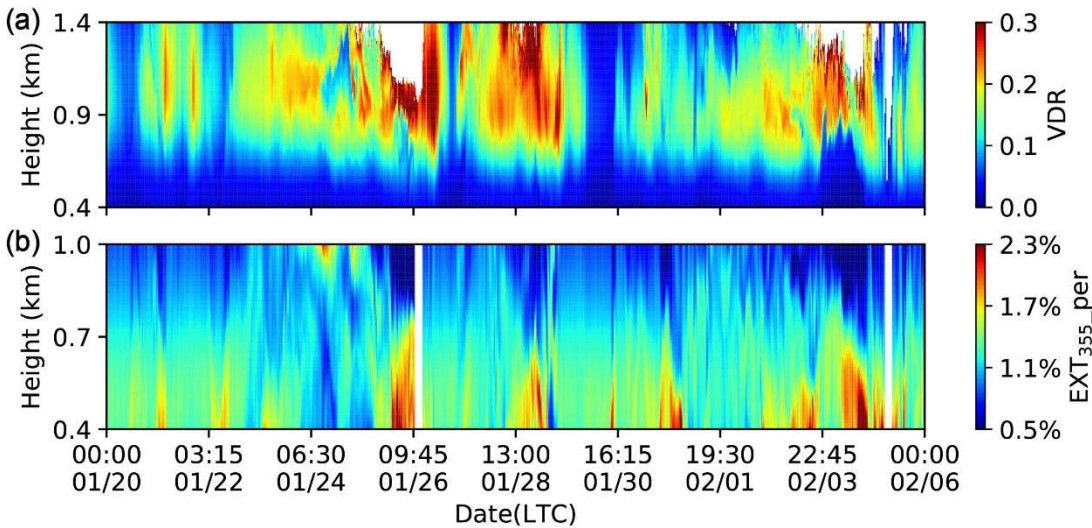

**Figure 7.** Correlation between VDR and $EXT_{355}$ from 20 Jan to 5 Feb 2017. The vertical structure of (**a**) VDR, and (**b**) percentage of $EXT_{355}$ of total $EXT_{355}$. The percentage of $EXT_{355}$ is used to characterize the aerosol concentrations at different heights, which is defined as: $EXT_{355\_per}(z) = 100\% \times EXT_{355}(z)/\sum_{z=400}^{1000} EXT_{355}(z)$ . Where $EXT_{355\_per}$ is the percentage of $EXT_{355}$, z is the height.

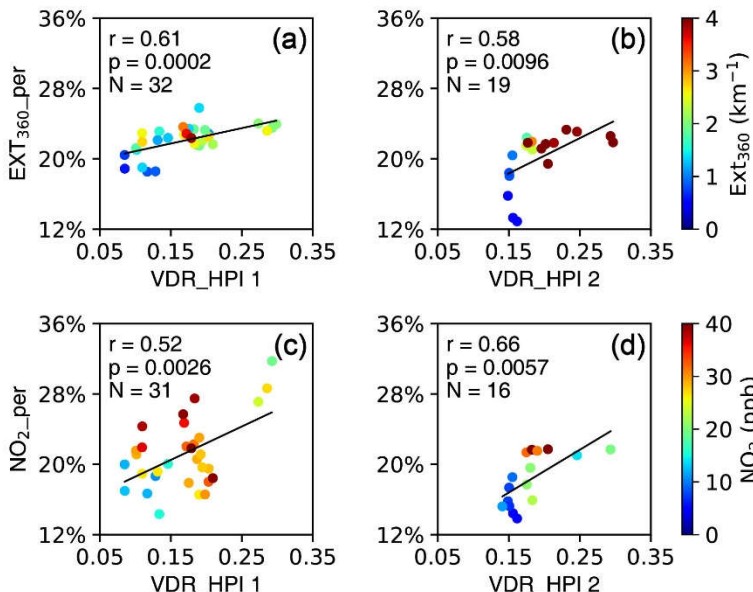

**Figure 8.** Average VDR trend in the upper lidar layer during the two HPIs and its impact on lower–level air pollution. Scatterplots showing relationships between the average VDR and the average percentages of bottom (**a**) $EXT_{360}$ during the HPI 1, (**b**) $EXT_{360}$ during the HPI 2, (**c**) $NO_2$ VMRs during the HPI 1, and (**d**) $NO_2$ VMRs during the HPI 2. The colors in (**a**) and (**b**) represent the surface $EXT_{360}$, and the colors in (**c**) and (**d**) represent the surface $NO_2$ concentration. The spatially averaged range of VDR is 950–1,050 m. The hourly averages of VDR, percentage of bottom $EXT_{360}$, and percentage of bottom $NO_2$ VMR are used due to the different temporal resolution between RL and MAX–DOAS. The correlation coefficients are shown at the top left, N=number of samples. The analysis period was from 8:00–16:00 LT because MAX–DOAS can only be performed during the day.

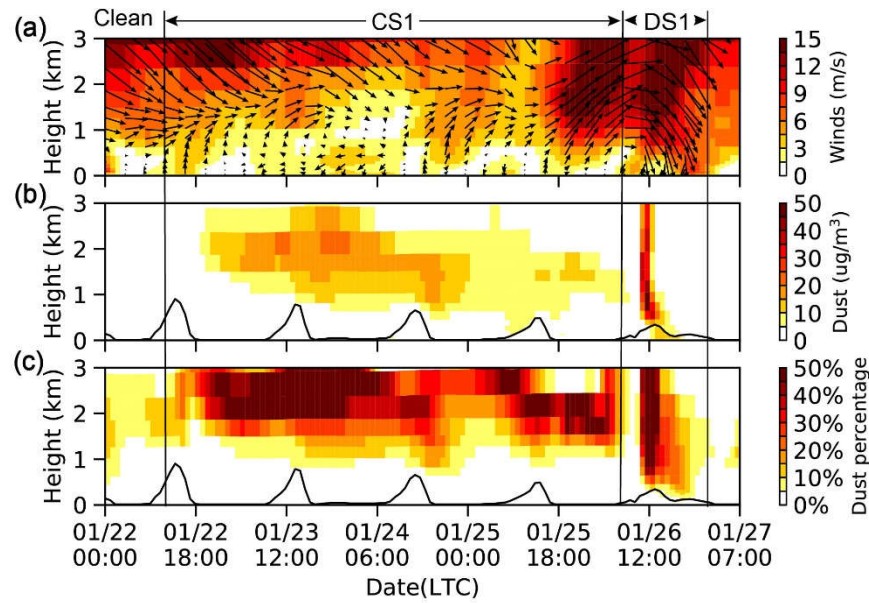

**Figure 9.** Curtain plots of winds and dust concentrations in CWBF from 22 to 26 Jan 2017. (**a**) Winds simulated by WRF–Chem, (**b**) vertical structure of dust concentrations, and (**c**) vertical structure of the composition of dust in total $PM_{10}$ concentrations. The black arrow in (**a**) indicates the wind direction, upper arrow for south winds. The black lines in (**b**) and (**c**) represent the PBL height evolution in WRF–Chem.

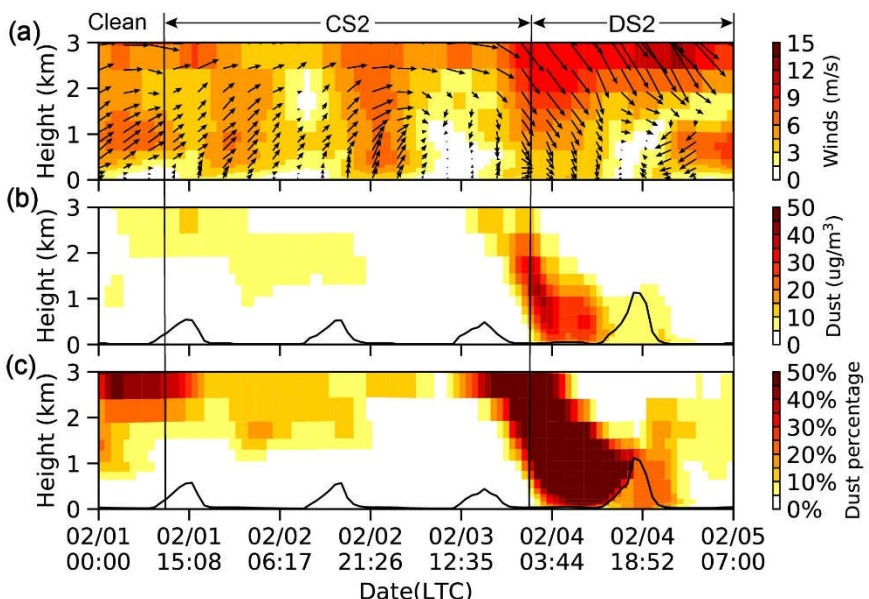

**Figure 10.** Same as Fig. 9 but from 1 to 5 Feb 2017.

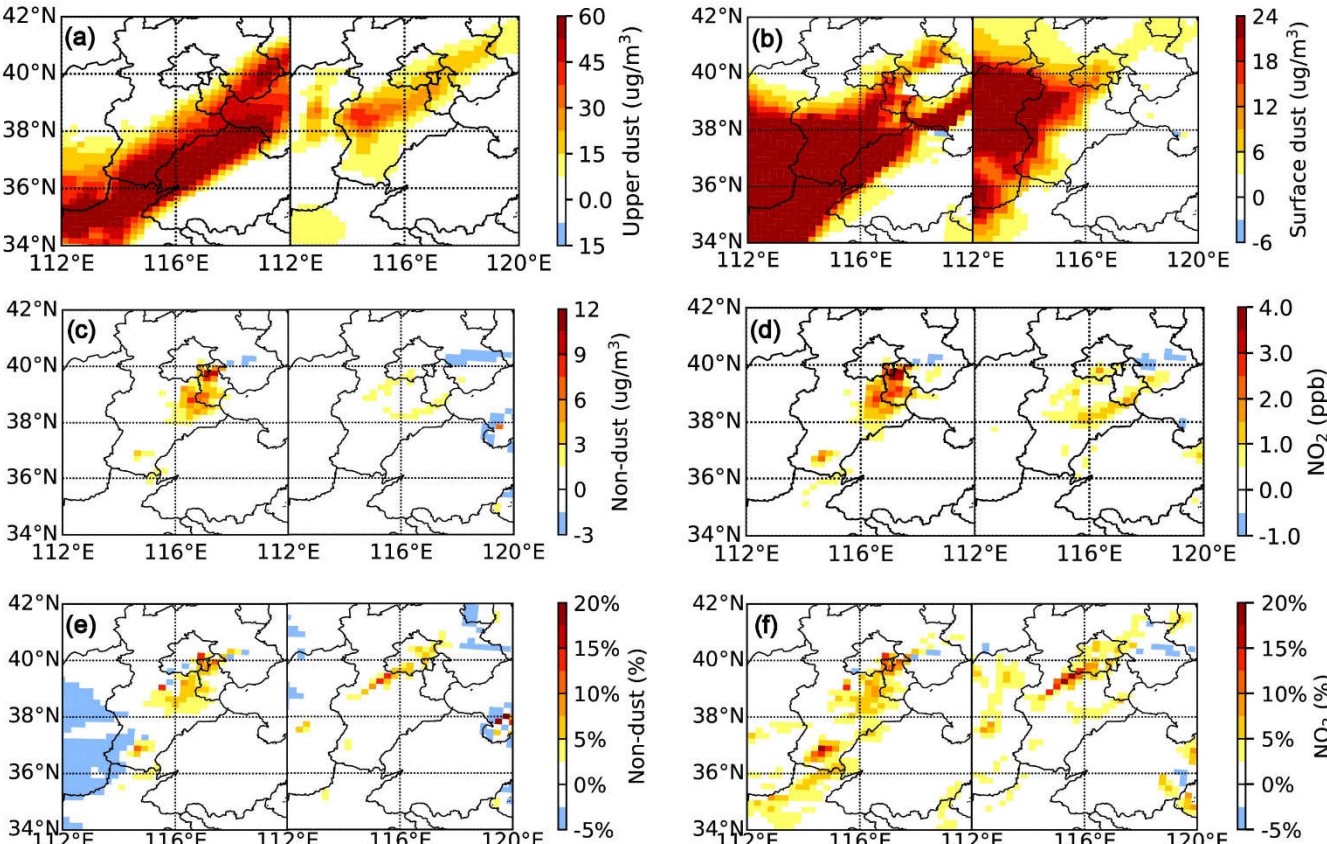

**Figure 11.** Influence of upper–level dust on surface non–dust aerosol in HPI 1 and HPI 2 dissipation stages. Horizontal distribution of (**a**) upper–level suspended dust concentration and (**b**) surface dust concentration. Difference in (**c**) surface non–dust particle concentration, (**d**) surface $NO_2$ concentration between the experiments dust_on and dust_off. The percentage change of (**e**) surface non–dust particle concentration and (**f**) surface $NO_2$ concentration between the experiments dust_on and dust_off. The time of each subgraph is the HPI 1 dissipation stage at 13:00 LT on 26 Jan 2017 (left panel) and HPI 2 dissipation stage at 16:00 LT on 4 Feb 2017 (right panel).

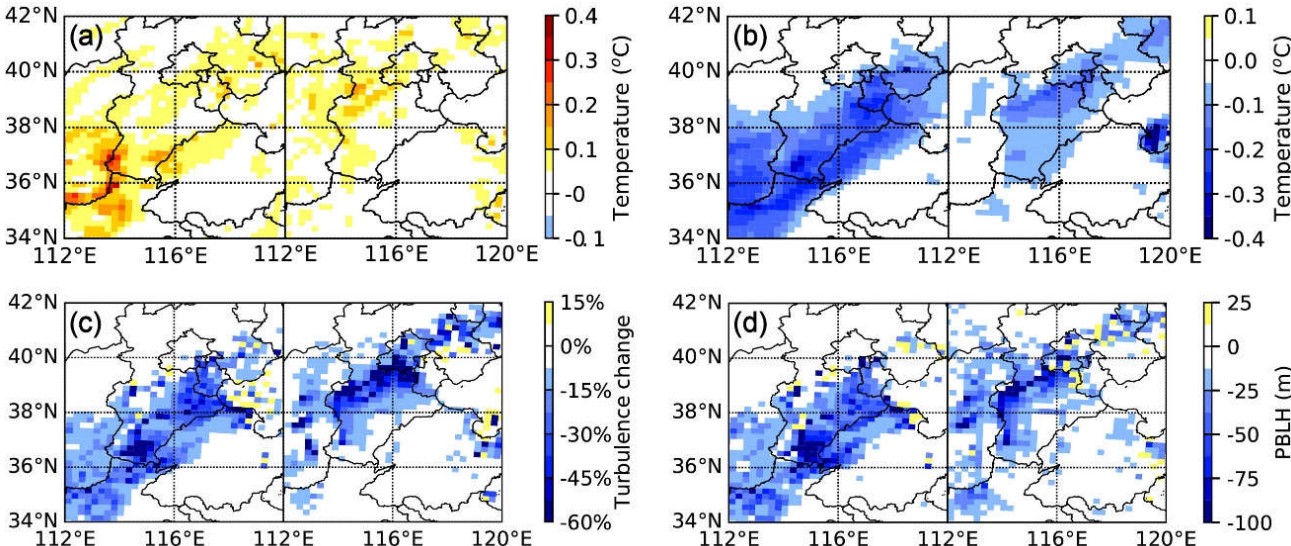

**Figure 12.** Influence of upper–level dust on meteorology parameters in HPI 1 and HPI 2 dissipation stages. (**a**) Maximum enhancement in temperature above the PBL, (**b**) Maximum temperature reduction within the PBL. Difference in (**c**) turbulence within PBL, and (**d**) PBL height between the experiments dust_on and dust_off. The time of each subgraph is the HPI 1 dissipation stage at 13:00 LT on 26 Jan 2017 (left panel) and HPI 2 dissipation stage at 16:00 LT on 4 Feb 2017 (right panel).

**Table 1.** Characteristics of the RL system.

| Transmitter | |
|---|---|
| Laser type | ND:YAG (QSmart850) |
| Wavelength (nm) | 355, 532 |
| Energy/pulse (mJ) | 230, 300 |
| Pulse repetition (Hz) | 10 |
| Beam divergence (mrad) | 0.5 |
| **Receiver** | |
| Collector | LICEL TR20–160 |
| Telescope type | Cassegrain |
| Field of view (mrad) | 0.2 |
| Telescope diameter (mm) | 400 |
| PMT | 532P (R9880–110) |
| | 532S (R9880–110) |
| | 355 (R9880–113) |
| | 387 (R9880–113) |
| | 408 (R9880–113) |
| Signal detection | Analog mode/Photo counting |
| Range resolution (m) | 7.5 |
| **Detected species** | |
| Mie/Rayleigh | 355 |
| Raman nitrogen | 387 |
| Raman water vapor | 408 |
| Polarization | 532p, 532s |
| **[1]Inputs of RL parameters** | |
| $EXT_{355}$ | $P_{355}(z)$ |
| $EXT_{532}$ | $P_{532}(z)$ |
| VDR | $P_{532p}(z)$, $P_{532s}(z)$ |
| LR | $P_{355}(z)$, $P_{387}(z)$ |
| Water vapor | $P_{355}(z)$, $P_{387}(z)$, $P_{408}(z)$ |

[1]Inputs are the elastic or inelastic backscatter signal profiles of RL. The subscript indicates the wavelength. The total elastic backscatter signal (Freudenthaler et al., 2009) profiles at 532 nm is defined as $P_{532}(z)=P_{532p}(z)+P_{532s}(z)$ .

**Table 2.** WRF-Chem model Configuration options.

| Configuration options | |
| --- | --- |
| Long-wave radiation | RRTMG |
| Short-wave radiation | RRTMG |
| Cumulus parameterization | Grell–Deveny (Grell and Dévényi, 2002) |
| Land-surface | Noah (Ek et al., 2003) |
| PBL | YSU (Mlawer et al., 1997) |
| Microphysics | Lin et al. (Lin et al., 1983) |
| Gas chemistry | CBMZ |
| Aerosol chemistry | MOSAIC (Zaveri et al., 2008) |

35

**Table 3.** RL parameters for Asian dust, ice clouds and anthropogenic aerosols.

| $S_\lambda$ (sr) | δ (%) | Location | Height (km) | Reference |
|---|---|---|---|---|
| **Asian dust** | | | | |
| [1]47±18($S_{532}$) | [2]20±7($\delta^p$) | Tsukuba, Japan | >5 | Sakai et al., 2003 |
| 46±5($S_{532}$) | 20–33($\delta^p$) | Tsukuba, Japan | 4–7 | Sakai et al., 2002 |
| 42–55($S_{532}$) | | Tsukuba, Japan | >2.5 | Liu et al., 2002 |
| 48.6±8.5($S_{355}$) | ~20($\delta^p$) | Tokyo, Japan | 3.5–4.3 | Murayama et al., 2004 |
| 46.5±10.5($S_{532}$) | ~30($\delta^p$) | Tokyo, Japan | 4.5–6.5 | Murayama et al., 2003 |
| 35±5($S_{532}$) | | Beijing, China | [4]PBL | Müller et al., 2007 |
| 36.2±4.7($S_{532}$) | [3]19.5±0.5($\delta^v$) | Beijing, China | 0.2–1.2 | Xie et al., 2008 |
| 40±5($S_{532}$) | 20–25($\delta^p$) | Beijing, China | 0.75–2.5 | Tesche et al., 2007 |
| **Ice clouds** | | | | |
| 10±30($S_{355}$) | 13–35($\delta^p$) | Arctic | >4 | Reichardt et al., 2002 |
| 29±12($S_{532}$) | 20–60($\delta^p$) | Chung–Li, Taiwan | ~12 | Chen et al., 2002 |
| ~20($S_{532}$) | | North America | >5 | Burton et al., 2012 |
| ~20($S_{355}$) | | Germany | >8 | Ansmann et al., 1992 |
| 17±14($S_{532}$) | 22±7($\delta^p$) | Tsukuba, Japan | >7 | Sakai et al., 2003 |
| 25±1($S_{355}$) | | Beijing, China | ~13 | Tao et al., 2012 |
| **Anthropogenic aerosols** | | | | |
| 56±6($S_{532}$) | 6±1($\delta^p$) | Central Europe | | Groß et al., 2013 |
| 50–70($S_{532}$) | <10 | North America | | Burton et al., 2012 |
| 73.9±6($S_{532}$) | 5.6±0.5($\delta^v$) | Beijing, China | | Xie et al., 2008 |
| 38.5±5($S_{532}$) | 7.2±1.4($\delta^v$) | Beijing, China | | Xie et al., 2008 |
| 60–70($S_{532}$) | | [5]GAW | | Hänel et al., 2012 |
| ~60($S_{532}$) | | Pearl River Delta | | Müller et al., 2006 |

[1]± for one standard deviation.
[2]The superscript "p" for linear particle depolarization ratio at 532 nm.
[3]The superscript "v" for linear volume depolarization ratio at 532 nm.
[4]PBL indicates the lower aerosol layer.
[5]Global Atmospheric Watch (GAW) station of Shangdianzi (46°N, 117°E) in the North China Plain 100 km northeast of Beijing.

**Table 4.** Stages of HPI along with the horizontal surface wind speed during the different pollution stages.

|  | Stage | Period (LTC) | [1]Wind speed |
|---|---|---|---|
| HPI 1 | CS1 | 2017/01/22 11:00–2017/01/26 05:00 | 1.8 |
|  | DS1 | 2017/01/26 06:00–2017/01/26 23:00 | 4.5 |
| HPI 2 | CS2 | 2017/02/01 11:00–2017/02/03 22:00 | 2.4 |
|  | DS2 | 2017/02/03 23:00–2017/02/05 06:00 | 2.8 |

[1] Spatially averaged wind speed below 100 m. Units: m/s.

