# Peer review of "Elevated dust layers inhibit dissipation of heavy anthropogenic surface air pollution"

_Atmospheric Chemistry and Physics, 2020_

## Referee Comment (RC1) · Anonymous Referee #1 · 19 Jun 2020

Title: Elevated dust layers inhibit dissipation of heavy anthropogenic surface air pollution MS No.: acp-2020-379

Comments: The paper explains mechanism of the elevated dust layer enhances surface air pollution during persistent heavy air pollution events. This paper is thought to be helpful in understanding the phenomenon of high concentration of fine particle (PM2.5) in winter and spring in Northeast Asia. The dataset presented is interesting and surely deserves publication. However, I am afraid the paper cannot be published as it is, as a number of points must be clarified. A detailed review follows:

Major Comments 1. The observation period of this paper is as short as 2 weeks. It is true that it produced important results for the generation and diffusion of yellow dust and pollutants, but it seems that it is difficult to generalize the results of the study due

to the limited number of analyzes with a short observation period. Therefore, I hope there is an expression that the paper can be applied under special conditions.

2. In order to understand the overall content of the paper, it seems necessary to check the supplementary materials. Including some of the materials in the supporting materials directly in the paper seems to be more helpful in understanding the paper. In particular, Figure S2 should be included in the next of the Figure 4 in the paper. Figure S3 also should be included in the paper.

3. Figure S5. In Figure S5, Data comparison of RL and MAX-DOAS is shown. But, just shown as correlation plot. Since the paper indicates that MAX-DOAS can be observed at different altitudes with a resolution of 100 m, it would be wise to show a graph that is compared with a profile that includes altitude distribution of RL and MAX-DOAS instead of a correlation plot like Figure S5 (c)and (d). Also, Figure S5 should be included the paper in "2 Measurements and methodology" part.

4. Suggestion: Figure 2(a,b,d) and Figure 3, 4(b,c,d) are overlapping. It would be nice to remove Figures 3 and 4 and express them as one in Figure 2.

Technical comments; 1. Page 5 line 5: Please include explain of "VMR".
* * *

---

## Referee Comment (RC2) · Anonymous Referee #2 · 25 Aug 2020

Comments on "Elevated dust layers inhibit dissipation of heavy anthropogenic surface air pollution" by Wang et al.

Severe winter haze is one of the major environmental issues in China. This is a comprehensive study combining remote sensing observations with three dimensional chemical transport model, which investigated the effect of natural dust on haze formation. The results show that a dust layer frequently exists above the planetary boundary layer (PBL), inhibits dilution of surface pollutants, and aggravates haze pollution. This study provides very important information toward a complete understanding of the formation mechanism of winter haze in China. The impact of dust is not limited to aerosol-PBL interactions, but may also explain the special multiphase chemistry in this region. Overall, this is a very nice study and I'd recommend its publication in ACP after minor revisions.

[Figure]

Major comments:

Snap shot vs full picture

In this version, the authors focused on a few pollution episodes. My question is how representative are these episodes. Since you have collected several month measurement data, I'd suggest you to provide more statistics. For example, are these dust layers always accompanying the haze events, if not, how frequent? How about the rest periods, is there still a dust layer, and what's the mean dust concentration in both bases? The authors mentioned "This stratification is governed by meteorological conditions that strong northwesterly winds usually prevailed in the lower free troposphere, and southerly winds are dominated in the PBL, producing persistent and intense haze pollution." How often do you have such meteorological conditions? Is there any episode with southerly winds both in the PBL and in the free troposphere?

Other comments:

Abstract "here we found that aerosols in North China are typically characterized by a pronounced vertical stratification ..." By saying "typically", do you mean in winter or all seasons?

Abstract "With the accumulation of elevated dust, the proportion of aerosol and trace gas at the surface in the whole column increased." Normally, we talked about accumulation of pollutants only near the surface referrring to the addition of emitted/secondary formation pollutants. For elevated dust (no emission and secondary formation), can you explain how it can be accumulated?

Page 2 line 6 "Accumulation of air pollutants from stationary and transportation sources, accompanied by the explosive increase of new particles under stagnant weather conditions (Guo et al., 2014; Huang et al., 2014), cause PM2.5 (particle mass less than 2.5 $\mu$m in diameter) concentrations to increase several–fold within a few hours." The explosive increase is not caused by accumulation by the transport

(Zheng et al. 2015). The new particles normally refer to sub-10 nm particles, while during severe haze event particles are $\sim$ 100 nm. The multiphase chemical formation (Cheng et al. 2016) is also an important pathway for the haze formation and should be included here.

Equation 3 "... and OIN are nitrate, sulfate, ammonium ..", What of nitrate? Concentration, mass, or?

Equation 4, "1. without considering the influence of dust (dust_off), that is, the effects of dust on radiation transfer and meteorology were ignored; 2. with consideration of the effect of dust (dust_on)," It is not clear how the two numerical experiments were carried out. According to these descriptions, the dust_off case is performed without considering the effects of dust on radiation and meteorology. But without considering dust and without considering the effect of dust are different. Also it is not clear how comes equation 4 because the difference between these two OINs may also be caused by the feedbacks on meteorological conditions on OINs other than dust. Why you cannot directly calculate dust composition from your model?

Page 6 equation 7 and 8, here you calculated the change of turbulence exchange coefficient, how about convection/advection, which is also important for pollutant transport?

Page 7 "had peak mass concentrations greater than 500 $\mu$g m-3." Mass concentration of what? Dust, PM2.5?

Page 8 line 7"The average EXT355 in the upper lidar layer during the weak southerly wind conditions was 1.00 km-1, which is clearly higher than that during the winds from Gobi 15 desert (0.66 km-1) and sparsely populated northern mountain areas (0.38 km-1)." Could you explain why during southerly wind conditions, the EXT355 in the upper layer is even higher? Since the other pollutants lead to a thicker/high abundance layer, will it have a stronger effect on the haze events, compared to the dust case (from Gobi desert and northern mountain areas)?

Page 8, line 17 "which is conducive to the accumulation and explosive growth of aerosols in the lower and upper lidar layers.", here again, there is no explosive growth due to chemical processes, and the apparent explosion is mainly caused by fast transition of air masses (Zheng et al. 2015).

Page 8 line 28 "During HPI 1, the upper dust layer formed slightly later than the accumulation of the anthropogenic aerosols in the lower lidar layer (Fig.3)." So?

Page 9 line 16, " The two parallel simulations, dust_on and dust_off, well reproduced the spatial and temporal variations of dust concentration at CWBF (Fig. 8 and Fig. 9)." How can dust_off well reproduced the dust concentration?

Section 3.3, I thought you may use dust_on and dust_off case to analyze the impact of dust, but the relevant discussion in the section is rather limited and vague. For example, "Consequently, dust–meteorology interactions result in more stagnant conditions, with the turbulent exchange coefficient within the PBL falling by over 60%. Similarly, a significant decrease in PBL height was also attributable to the stable stratification (Fig. 11c and 11d).", what did you define the dust-meteorology interactions? How did you calculate the change of PBL and turbulence? Based on comparison between different periods/stages or between the two scenarios (dust_on and dust_off)? The 60% reduction of turbulent exchange coefficient seems to be a large effect, but the change of NO2 and aerosol concentrations seem to be small. Can you also calculate the percentage change due to dust in analogy to the absolute change in Figure 10.

Reference:

Zheng, G. J., Duan, F. K., Su, H., Ma, Y. L., Cheng, Y., Zheng, B., Zhang, Q., Huang, T., Kimoto, T., Chang, D., Pöschl, U., Cheng, Y. F., and He, K. B.: Exploring the severe winter haze in Beijing: the impact of synoptic weather, regional transport and heterogeneous reactions, Atmos. Chem. Phys., 15, 2969-2983, 10.5194/acp-15-2969-2015, 2015.

[Figure]

Cheng, Y., Zheng, G., Wei, C., Mu, Q., Zheng, B., Wang, Z., Gao, M., Zhang, Q., He, K., Carmichael, G., Poschl, U., and Su, H.: Reactive nitrogen chemistry in aerosol water as a source of sulfate during haze events in China, Sci. Adv., 2, 10.1126/sciadv.1601530, 2016.

———————————————————

---

## Author Comment (AC1) · 27 Sep 2020

We truly grateful for the reviewers' positive assessments of our manuscript and the helpful suggestions. We have revised the manuscript carefully according to the reviewers' comments. Point-to-point responses are given below. The original comments are black in color, while our responses are in blue. The revised parts in the manuscript are marked in red. All the page number and line number are referred to the revised manuscript.

**Major comments**

(1)    The observation period of this paper is as short as 2 weeks. It is true that it produced important results for the generation and diffusion of yellow dust and pollutants, but it seems that it is difficult to generalize the results of the study due to the limited number of analyzes with a short observation period. Therefore, I hope there is an expression that the paper can be applied under special conditions.

R: We have further supplemented the observation data in the manuscript and supplementary material (Figure R1 and Figure S5 in the supplementary materials). Nine heavy pollution incidents (HPI) have been observed and 8 HPIs present aerosol stratification (except HPI 3), the duration of each case is listed in the Table R1 and Table S1 in the supplementary materials. The aerosol stratification is most prominent in HPI 1 and HPI 2, the VDR in the upper lidar layer during dissipation stage was greater than 0.3, suggesting almost pure dust. We have analyzed these two HPIs in detail in the manuscript. In addition, we describe the scope of application of this article in the summary section of the manuscript. When the southerly wind bringing anthropogenic aerosols was dominant in the planetary boundary layer and northwest wind bringing dust was prevailing in the free troposphere (dust), the stratification of aerosols occurred. Upper dust enhances temperature inversions, reduces PBL height, and suppresses convection, ultimately resulting in the increase of surface air pollutants. We supplemented these materials in the manuscript. Please refer to Page 8 Line 11–15 and Page 12 Line 20–30.

[Figure]

***Figure R1***. *Periodic air pollution cycles during our whole observation. The color contours show the vertical structure of **(a)** EXT$_{355}$ and **(b)** VDR. **(c)** Temporal evolutions of surface average PM$_{2.5}$ and PM$_{10}$ mass concentrations observed by six environmental monitoring stations in Baoding. Each HPI is marked with a red rectangle in **(a)**, and the HPI number is displayed on the top of each red rectangle. The detailed date of each HPI is listed in Table R1.*

***Table R1.*** *The duration of each HPI during our whole observation.*

| Case | Period (LTC) |
|------|--------------|
| HPI 1 | 2017/01/22 11:00–2017/01/26 23:00 |
| HPI 2 | 2017/02/01 11:00–2017/02/05 06:00 |
| HPI 3 | 2017/01/05 04:00–2017/01/08 04:00 |
| HPI 4 | 2017/01/15 10:00–2017/01/19 12:00 |
| HPI 5 | 2017/01/27 14:00–2017/01/29 07:00 |
| HPI 6 | 2017/02/14 16:00–2017/02/16 13:00 |
| HPI 7 | 2017/02/18 00:00–2017/02/19 18:00 |
| HPI 8 | 2017/03/03 10:00–2017/03/05 20:00 |
| HPI 9 | 2017/03/16 03:00–2017/03/23 05:00 |

(2) In order to understand the overall content of the paper, it seems necessary to check the supplementary materials. Including some of the materials in the supporting materials directly in the paper seems to be more helpful in understanding the paper. In particular, Figure S2 should be included in the next of the Figure 4 in the paper. Figure S3 also should be included in the paper.

R: We have carefully checked the content of the supplementary material and included the Figure S2, Figure S3 and some important descriptions in the manuscript.

(3)   Figure S5. In Figure S5, Data comparison of RL and MAX-DOAS is shown. But, just shown as correlation plot. Since the paper indicates that MAX-DOAS can be observed at different altitudes with a resolution of 100 m, it would be wise to show a graph that is compared with a profile that includes altitude distribution of RL and MAX-DOAS instead of a correlation plot like Figure S5 (c) and (d). Also, Figure S5 should be included the paper in "2 Measurements and methodology" part.

R: The comparison of mean aerosol extinction coefficient (EXT) profile during HPI 1 and HPI 2 between Raman lidar (RL) and MAX–DOAS was shown in Fig. R2 and Figure S2 in the supplementary materials. The EXT profile measured by RL usually greater than the EXT profile observed by RL. The correlation of hourly and spatially average EXT from 400 m to 600 m and 600 m to 800 m between RL and MAX–DOAS show a reasonably good agreement (R > 0.8), while the slope of linear regression between RL and MAX–DOAS measured EXT is considerably less than 1 (Fig. 2 in the manuscript). Because the sensitivity of the MAX–DOAS measurements decreases with increasing altitude in the troposphere (Frieß et al., 2006). Thus, only the EXT profiles below 800 m measured by MAX–DOAS were used in the manuscript. In addition, MAX–DOAS and lidar measurements were made with different geometries (a combination of zenith–sky and off–axis versus zenith–sky only, respectively) and different integration times for completing a set of measurements (15 versus 22 min, respectively), which may also explain part of the differences between the EXT profiles measured by RL and MAX–DOAS (Irie et al., 2008).

In addition, we have also supplemented the Figure R2 in the supplementary materials and included the Figure S5 and some important descriptions in the manuscript.

[Figure]

***Figure R2.*** *Comparison of average EXT profile during HPI 1 (left) and HPI 2 (right) between RL and MAX–DOAS.*

(4)  Suggestion: Figure 2 (a, b, d) and Figure 3, 4 (b, c, d) are overlapping. It would be nice to remove Figures 3 and 4 and express them as one in Figure 2.

R: We have followed this suggestion and express them as one in Figure 3 in the manuscript.

**Technical comments**

①  Page 5 line 5: Please include explain of "VMR".

R: We have followed this suggestion and corrected the mistake accordingly.

**Reference:**

Draxler, R. R., and Hess, G.: An overview of the HYSPLIT_4 modelling system for jectories, Aust. Meteorol. Mag., 47, 295-308, 1998.

Frieß, U., Monks, P. S., Remedios, J. J., Rozanov, A., Sinreich, R., Wagner, T., and Platt, U.: MAX-DOAS O4 measurements: A new technique to derive information on atmospheric aerosols: 2. Modeling studies, J. Geophys. Res., 111, D14203 doi: 10.1029/2005jd006618, 2006.

Irie, H., Kanaya, Y., Akimoto, H., Iwabuchi, H., Shimizu, A., and Aoki, K.: First retrieval of tropospheric aerosol profiles using MAX-DOAS and comparison with lidar and sky radiometer measurements, Atmos. Chem. Phys., 8, 341-350, doi: 10.5194/acp-8-341-2008, 2008.

---

## Author Comment (AC2) · 27 Sep 2020

We really appreciate the reviewers for the valuable and constructive comments, which are very useful for the improvement of the manuscript. We have replied the reviewers' comments point-to-point in below. The reviewers' comments are cited in black, while the responses are in blue. The revised parts in the manuscript are marked in red. All the page number and line number are referred to the revised manuscript.

**Major comments**

(1)   In this version, the authors focused on a few pollution episodes. My question is how representative are these episodes. Since you have collected several month measurement data, I'd suggest you to provide more statistics. For example, are these dust layers always accompanying the haze events, if not, how frequent? How about the rest periods, is there still a dust layer, and what's the mean dust concentration in both bases? The authors mentioned "This stratification is governed by meteorological conditions that strong northwesterly winds usually prevailed in the lower free troposphere, and southerly winds are dominated in the PBL, producing persistent and intense haze pollution." How often do you have such meteorological conditions? Is there any episode with southerly winds both in the PBL and in the free troposphere?

R: Periodic air pollution cycles during our whole observation is shown in Figure R1 and Figure S5 in the supplementary materials. Nine heavy pollution incidents (HPI) have been observed and 8 HPIs present aerosol stratification (except HPI 3), the duration of each case is listed in the Table R1 and Table S1 in the supplementary materials. The aerosol stratification is most prominent in HPI 1 and HPI 2, the VDR in the upper lidar layer during dissipation stage was greater than 0.3, suggesting almost pure dust. We have analyzed these two HPIs in detail in the manuscript. Among the eight HPIs where aerosol stratification occurred, the upper dust layer is strongly affected by the northwest transmission while the lower anthropogenic aerosols usually related to the southerly transportation (Figure R2 and Figure S6 in the supplementary materials). The upper dust layer does not always last the entire lower anthropogenic aerosol pollution period, such as HPI 9. Similarly, when there is no lower anthropogenic aerosol pollution, there will also be dust layer in the upper lidar layer (Figure R1b white

rectangle). During HPI 1 and HPI 2, the WRF-Chem simulation results show that the concentrations of elevated dust is 0–165μg/m$^3$ and 0–79 μg/m$^3$, respectively. We also found an episode (HPI 3) with southerly (southeast or southwest) winds both in the PBL and in the free troposphere (Figure R2 and Figure S6 in the supplementary materials). The VDR during HPI 3 in the lower and upper lidar layer is less than 0.08, indicating anthropogenic aerosols.

We also include these important statistics message in the manuscript. Please refer to Page 8 Line 11–15 and Page 12 Line 20–30.

[Figure]

***Figure R1***. *Periodic air pollution cycles during our whole observation. The color contours show the vertical structure of **(a)** EXT$_{355}$ and **(b)** VDR. **(c)** Temporal evolutions of surface average PM$_{2.5}$ and PM$_{10}$ mass concentrations observed by six environmental monitoring stations in Baoding. Each HPI is marked with a red rectangle in **(a)**, and the HPI number is displayed on the top of each red rectangle. The detailed date of each HPI is listed in Table R1.*

[Figure]

***Figure R2.*** *Cluster analysis of 24–h air mass backward trajectories (AMBTs) initialized at 500 m (black) and 1000 m (red) during each HPI. The numbers in the map are the fraction of each category of AMBTs. The 24–hour AMBTs were computed using the Hybrid Single–Particle Lagrangian Integrated Trajectory (HYSPLIT) model of the National Oceanic and Atmospheric Administration (Draxler and Hess, 1998). We calculated the hourly AMBTs during the whole observation period initialized at 500 m and 1000 m. Then, cluster analysis of AMBTs was conducted in two categories directions. The HPI number is shown in top left of each panel.*

***Table R1.*** *The duration of each HPI during our whole observation.*

| Case | Period (LTC) |
| --- | --- |
| HPI 1 | 2017/01/22 11:00–2017/01/26 23:00 |
| HPI 2 | 2017/02/01 11:00–2017/02/05 06:00 |
| HPI 3 | 2017/01/05 04:00–2017/01/08 04:00 |
| HPI 4 | 2017/01/15 10:00–2017/01/19 12:00 |
| HPI 5 | 2017/01/27 14:00–2017/01/29 07:00 |
| HPI 6 | 2017/02/14 16:00–2017/02/16 13:00 |
| HPI 7 | 2017/02/18 00:00–2017/02/19 18:00 |
| HPI 8 | 2017/03/03 10:00–2017/03/05 20:00 |
| HPI 9 | 2017/03/16 03:00–2017/03/23 05:00 |

**Other comments**

(1) Abstract "here we found that aerosols in North China are typically characterized by a pronounced vertical stratification ..." By saying "typically", do you mean in winter or all seasons?

R: Here we refer to winter, because we only analyzed the vertical distribution of winter–time aerosols, we corrected the description as "here we found that winter–time aerosols in North China are typically characterized by a pronounced vertical stratification...". Please refer to Page 1 Line 27.

(2) Abstract "With the accumulation of elevated dust, the proportion of aerosol and trace gas at the surface in the whole column increased." Normally, we talked about accumulation of pollutants only near the surface referring to the addition of emitted/secondary formation pollutants. For elevated dust (no emission and secondary formation), can you explain how it can be accumulated?

R: Thanks for pointing out the unsuitable expression. Here we mean the proportion of dust aerosol concentrations in total aerosol concentrations has increased. Because the air pollution in the upper lidar layer is affected by the northwest transport, the VDR in the upper lidar layer increases continuously, indicating that the contributions of dust to total aerosol concentrations has increased. We have re-phrased the sentence as "With the increased contribution of elevated dust to the upper aerosols". Please refer to Page 1 Line 32.

(3) Page 2 line 6 "Accumulation of air pollutants from stationary and transportation sources, accompanied by the explosive increase of new particles under stagnant weather conditions (Guo et al., 2014; Huang et al., 2014), cause $PM_{2.5}$ (particle mass less than 2.5 $\mu$m in diameter) concentrations to increase several–fold within a few hours." The explosive increase is not caused by accumulation by the transport (Zheng et al. 2015). The new particles normally refer to sub-10 nm particles, while during severe haze event particles are $\sim$ 100 nm. The multiphase chemical formation (Cheng et al. 2016) is also an important pathway for the haze formation and should be included here.

R: We have included the multiphase chemical formation pathway for the haze formation and cited the related references in the manuscript. We have improved description as "Accumulation of air pollutants from stationary and transportation sources and explosive increase of new particles under stagnant weather conditions (Guo et al., 2014; Huang et al., 2014; Zheng et al. 2015) through chemical reaction, such as multiphase chemical formation (Cheng et al. 2016) as well as regional transport (Li et al., 2017), cause $PM_{2.5}$ (particle mass less than 2.5 μm in diameter) concentrations to increase several–fold within a few hours.". This information is now included in the manuscript (Page 2 Line 8–12).

(4) Equation 3 "... and OIN are nitrate, sulfate, ammonium ..", What of nitrate? Concentration, mass, or?

R: The $NO_3$, $SO_4$, $NH_4$, OC, BC, CL, NA, and OIN are the 3–D mass mixing ratios of the MOSAIC variables in the MOSAIC aerosol scheme, the unit of 3–D mass mixing ratios is μg/kg. We corrected the description of these variables as "the $NO_3$, $SO_4$, $NH_4$, OC, BC, CL, NA, and OIN are 3–D mass mixing ratios of nitrate, sulfate, ammonium, organic compounds, black carbon, chloride, sodium, and other inorganic compounds, respectively.". Please refer to Page 5 Line 29–30.

(5) Equation 4, "1. without considering the influence of dust (dust_off), that is, the effects of dust on radiation transfer and meteorology were ignored; 2. with consideration of the effect of dust (dust_on)," It is not clear how the two numerical experiments were carried out. According to these descriptions, the dust_off case is performed without considering the effects of dust on radiation and meteorology. But without considering dust and without considering the effect of dust are different. Also it is not clear how comes equation 4 because the difference between these two OINs may also be caused by the feedbacks on meteorological conditions on OINs other than dust. Why you cannot directly calculate dust composition from your model?

R: Thanks for pointing out the unsuitable expression. We turned off the dust emission in our simulation area in dust_off case, indicating without considering dust. The influence of elevated dust on meteorological conditions mainly includes two aspects.

One is the influence of elevated dust itself on radiation. Secondly, the elevated dust also promotes chemical reactions and the formation of new particles in the upper layer (Cwiertny et al., 2008; Nie et al., 2014), and the newly formed upper aerosols induced by elevated dust can also affect the radiation. Here we re-phrased the sentence as "1. without considering the dust (dust_off); 2. with consideration of the dust (dust_on).", please refer to page 6 line 2 and page 12 line 2–3.

Our WRF–Chem model is public version 3.6, and we uses the CBMZ/MOSAIC chemical mechanism (Zaveri et al., 2008), which does not identify dust as a separate species. The emitted dust is assigned to the other inorganic compounds (OIN) class of MOSAIC (Zaveri et al., 2008). Indeed, the feedbacks on meteorological conditions on OINs will cause the OIN differences between the two scenarios (dust_on and dust_off). However, according to the effects of elevated dust on non–dust particles (nitrate, sulfate, ammonium, organic compounds, and black carbon), elevated dust has increased the surface non–dust particles by 0%–21%, while the elevated dust has insignificant effects on upper non–dust particles (less than 5%). By analogy, the difference in OIN at the surface between the two experiments should also be increased by 0%–21% (0–23 $\mu g/m^3$) due to elevated dust, and OIN in the upper layer should be almost unchanged. Actually, the difference in surface OIN between the two experiments has increased up to 109 $\mu g/m^3$ and the difference in upper OIN has increased up to 165 $\mu g/m^3$, indicating approximately 80% surface OIN difference and almost all the upper OIN difference were caused by dust. Thus, we approximately calculated the dust concentration as the OIN difference between the two scenarios (dust_on and dust_off), which is expressed in Equation 4 in the manuscript.

(6)   Page 6 equation 7 and 8, here you calculated the change of turbulence exchange coefficient, how about convection/advection, which is also important for pollutant transport?

R: Thanks for the suggestion. In the revised manuscript, we use horizontal winds to indicate advection and convective precipitation to reflect convection, and lower convective precipitation suggests weaker convection (Baro et al., 2015; Gao et al.,

2013). We calculate the change of surface horizontal winds speed between two experiments (dust_on and dust_off). The maximum reduction of surface horizontal winds speed up to 1.2 m/s, the relative attenuation of surface horizontal winds speed is 0%–27% (Figure R3 and Figure S7 in the supplementary materials). Therefore, the elevated dust also weakens the surface advection. For convection, the average convective precipitation (RAINC) from 20 Jan to 4 Feb 2017 between two experiments (dust_on and dust_off) are extremely small, which implies that there is no active convection activity (Figure R3 and Figure S7 in the supplementary materials) during our observed period. The discussion about the advection and convection was also added in the manuscript. Please refer to Page 11 Line 16–19.

[Figure]

***Figure R3.*** *Influence of elevated dust on surface winds in HPI 1 and HPI 2 dissipation stages. (a) Difference in surface horizontal winds between the experiments dust_on and dust_off. (b) Percentage change in surface horizontal winds between the experiments dust_on and dust_off. The time of each subgraph is the HPI 1 dissipation stage at 13:00 LT on 26 Jan 2017 (left panel) and HPI 2 dissipation stage at 16:00 LT on 4 Feb 2017 (right panel). (c) Mean convective precipitation (RAINC) between two experiments dust_on (left) and dust_off (right) in WRF-Chem simulations from 20 Jan to 4 Feb 2017*

(7)    Page 7 "had peak mass concentrations greater than 500 µg m$^{-3}$." Mass concentration of what? Dust, PM$_{2.5}$?

R: Here we refer to PM$_{2.5}$ mass concentration, we have corrected the description as "had peak PM$_{2.5}$ mass concentrations greater than 500 µg m$^{-3}$". Please refer to Page 8 Line 16.

(8)    Page 8 line 7"The average EXT$_{355}$ in the upper lidar layer during the weak southerly wind conditions was 1.00 km$^{-1}$, which is clearly higher than that during the winds from Gobi desert (0.66 km$^{-1}$) and sparsely populated northern mountain areas (0.38 km$^{-1}$)." Could you explain why during southerly wind conditions, the EXT$_{355}$ in the upper layer is even higher? Since the other pollutants lead to a thicker/high abundance layer, will it have a stronger effect on the haze events, compared to the dust case (from Gobi desert and northern mountain areas)?

R: The high EXT$_{355}$ in the upper layer is often affected by anthropogenic aerosols transported from the south, resulting in an increase of EXT$_{355}$ and a decrease in VDR (Figure R4 red rectangle). Recent research has found that, based on the blocking role of mountains, a vertical vortex in the lower troposphere was induced over downwind regions. This mountain–induced vortex elevated ground pollutants to higher layers and formed a thick pollutant layer from the surface to above 1 km. The elevated pollutant layer is then transported to Beijing via an enhanced southerly wind, leading to aerosol pollution in the upper air of Beijing (Quan et al., 2019). Elevated dust have a greater impact on surface aerosols than that of anthropogenic pollutants during HPI 2. We compared the percentage of bottom EXT$_{360}$ in total EXT$_{360}$ on 23 and 24 Jan. The dust dominants the upper aerosols between 12:00–16:00 on 23 Jan, while between 12:00–16:00 on Jan 24, anthropogenic pollutants appeared in upper aerosols (Figure R4 red rectangle). The average percentage of bottom EXT$_{360}$ is 22.3% between 12:00–16:00 on 23 Jan and 21.2% between 12:00–16:00 on 24 Jan.

[Figure]

***Figure R4.*** *Periodic air pollution cycles in North China. The color contours show the vertical structure of (a) $EXT_{355}$ and (b) VDR measured by RL. Temporal evolutions of spatially average VDR and $EXT_{355}$ at (c) 950 m–1,050 m. The colors in (c) represent the air masses originating from southwest (red), Gobi desert (yellow), and sparsely populated northern moutain areas (blue).*

(9)    Page 8, line 17 "which is conducive to the accumulation and explosive growth of aerosols in the lower and upper lidar layers.", here again, there is no explosive growth due to chemical processes, and the apparent explosion is mainly caused by fast transition of air masses (Zheng et al. 2015).

R: We have re-phrased the sentence as "The shift of the origin of the air mass from northerly to southerly, together with a considerable decrease in wind speed, promotes the southerly transport of industrial pollutants and explosive increase of new particles under stagnant weather conditions (Zheng et al. 2015) through chemical reaction, such as multiphase chemical formation (Cheng et al. 2016), which is conducive to the accumulation of aerosols in the lower and upper lidar layers.". Please refer to Page 9 Line 19–21.

(10)    Page 8 line 28 "During HPI 1, the upper dust layer formed slightly later than the

accumulation of the anthropogenic aerosols in the lower lidar layer (Fig.3)." So?

R: As described in our manuscript, the low–level anthropogenic aerosols came from the southerly polluted industrial regions and the upper dust layers arrived mostly from Mongolia. The upper dust layer formed slightly later than the anthropogenic aerosols in the lower lidar layer during HPI 1, while the upper dust layer during HPI 2 appeared earlier than the anthropogenic aerosols in the lower lidar layer, indicating the formation of upper dust is independent of the formation of anthropogenic aerosols in the lower lidar layer. We have supplemented this information in Page 9 Line 32 and Page 10 Line 1 in the manuscript.

(11)   Page 9 line 16, "The two parallel simulations, dust_on and dust_off, well reproduced the spatial and temporal variations of dust concentration at CWBF (Fig. 8 and Fig. 9)." How can dust_off well reproduced the dust concentration?

R: Thanks for pointing out the unsuitable expression. The dust concentration was the OIN difference between the two scenarios (dust_on and dust_off), which is defined in Equation 4 in the manuscript. We have re-phrased the sentence as "The dust concentrations is derived from the OIN difference between the two scenarios of dust_on and dust_off (Equation 4), model simulations well reproduced the spatial and temporal variations of dust concentration at CWBF (Fig. 9 and Fig. 10)." . Please refer to Page 10 Line 20–21.

(12)   Section 3.3, I thought you may use dust_on and dust_off case to analyze the impact of dust, but the relevant discussion in the section is rather limited and vague. For example, "Consequently, dust–meteorology interactions result in more stagnant conditions, with the turbulent exchange coefficient within the PBL falling by over 60%. Similarly, a significant decrease in PBL height was also attributable to the stable stratification (Fig. 11c and 11d).", what did you define the dust-meteorology interactions? How did you calculate the change of PBL and turbulence? Based on comparison between different periods/stages or between the two scenarios (dust_on and dust_off)? The 60% reduction of turbulent exchange coefficient seems to be a large

effect, but the change of NO$_2$ and aerosol concentrations seem to be small. Can you also calculate the percentage change due to dust in analogy to the absolute change in Figure 10.

R: The dust-meteorology interactions mainly includes two aspects. Firstly, the difference in meteorological conditions between the upper and lower lidar layer leads to the aerosol stratification, dust or mixtures of dust and anthropogenic aerosols dominated above the PBL and anthropogenic aerosols prevailed within the PBL. Secondly, elevated dust alters the atmospheric thermodynamics and stability, mostly by lower–level cooling and upper–level heating, especially during dissipation stage. The suppressed turbulence exchange and decreased in PBL height impede dissipation of persistent heavy haze pollution. The change of PBL and turbulence was calculated between the two scenarios (dust_on and dust_off). The change of NO$_2$ and aerosol concentrations interact strongly with many meteorological variables, e.g. wind speed, temperature, humidity, turbulence, PBL (Li et al., 2017; Zhong et al., 2018). Model results show that, although the turbulent exchange coefficient decreased approximately 60%, the horizontal wind speed decreased 0–27% and the PBL decreased 0–26%. Thus, the deteriorating meteorological conditions resulted in the surface non–dust particle and NO$_2$ concentrations increased by 0%–21% (Figure R6 and Figure 11 in the manuscript). The concentration of surface non–dust particles and NO$_2$ increased by 0–11.4 µg/m3 and 0–4.4 ppb, respectively.

The definition of dust-meteorology interactions was added in the manuscript. Please refer to Page 12 Line 20–30. The percentage change of aerosol and NO$_2$ concentrations was also supplemented in the manuscript, please refer to Figure 11 and Page 11 Line 7–8.

[Figure]

**Figure R6.** *Influence of upper–level dust on surface non–dust aerosol in HPI 1 and HPI 2 dissipation stages. Horizontal distribution of (a) upper–level suspended dust concentration and (b) surface dust concentration. Difference in (c) surface non–dust particle concentration, (d) surface NO$_2$ concentration between the experiments dust_on and dust_off. The percentage change of (e) surface non–dust particle concentration and (f) surface NO$_2$ concentration between the experiments dust_on and dust_off. The time of each subgraph is the HPI 1 dissipation stage at 13:00 LT on 26 Jan 2017 (left panel) and HPI 2 dissipation stage at 16:00 LT on 4 Feb 2017 (right panel).*

**Reference:**

Baró, R., Jiménez-Guerrero, P., Balzarini, A., Curci, G., Forkel, R., Grell, G., & Pirovano, G. Sensitivity analysis of the microphysics scheme in WRF-Chem contributions to AQMEII phase 2. Atmos. Environ., 115, 620-629, doi: 10.1016/j.atmosenv.2015.01.047, 2015.

Cheng, Y., Zheng, G., Wei, C., Mu, Q., Zheng, B., Wang, Z., Gao, M., Zhang, Q., He, K., Carmichael, G., Poschl, U., and Su, H.: Reactive nitrogen chemistry in aerosol water as a source of sulfate during haze events in China, Sci. Adv., 2, 10.1126/sciadv.1601530, 2016.

Cwiertny, D. M., Young, M. A., and Grassian, V. H.: Chemistry and photochemistry of mineral dust aerosol, Annu. Rev. Phys. Chem., 59, 27-51, doi:

10.1146/annurev.physchem.59.032607.093630, 2008.

Draxler, R. R., and Hess, G.: An overview of the HYSPLIT_4 modelling system for trajectories, Aust. Meteorol. Mag., 47, 295-308, 1998.

Gao, Y., Wu, T., Chen, B., Wang, J., & Liu, Y. A numerical simulation of microphysical structure of cloud associated with the 2008 winter freezing rain over southern China. J. Meteor. Soc. Japan, 91, 101-117, 2013.

Li, Z., Guo, J., Ding, A., Liao, H., Liu, J., Sun, Y., Wang, T., Xue, H., Zhang, H., and Zhu, B.: Aerosol and boundary-layer interactions and impact on air quality, Natl. Sci. Rev., 4, 810-833, doi: 10.1093/nsr/nwx117, 2017.

Nie, W., Ding, A., Wang, T., Kerminen, V. M., George, C., Xue, L., Wang, W., Zhang, Q., Petaja, T., Qi, X., Gao, X., Wang, X., Yang, X., Fu, C., and Kulmala, M.: Polluted dust promotes new particle formation and growth, Sci. Rep., 4, 6634, doi: 10.1038/srep06634, 2014.

Quan, J., Dou, Y., Zhao, X., Liu, Q., Sun, Z., Pan, Y., Jia, X., Cheng, Z., Ma, P., Su, J., Xin, J., and Liu, Y.: Regional atmospheric pollutant transport mechanisms over the North China Plain driven by topography and planetary boundary layer processes, Atmos. Environ., doi: 10.1016/j.atmosenv.2019.117098, 2019.

Zaveri, R. A., Easter, R. C., Fast, J. D., Peters, L. K., Model for simulating aerosol interactions and chemistry (MOSAIC). J. Geophys. Res. 113, D13204, doi: 10.1029/2007jd008782, 2008.

Zheng, G. J., Duan, F. K., Su, H., Ma, Y. L., Cheng, Y., Zheng, B., Zhang, Q., Huang, T., Kimoto, T., Chang, D., Pöschl, U., Cheng, Y. F., and He, K. B.: Exploring the severe winter haze in Beijing: the impact of synoptic weather, regional transport and heterogeneous reactions, Atmos. Chem. Phys., 15, 2969-2983, 10.5194/acp-15-2969-2015, 2015.

Zhong, J., Zhang, X., Dong, Y., Wang, Y., Liu, C., Wang, J., Zhang, Y., and Che, H.: Feedback effects of boundary-layer meteorological factors on cumulative explosive growth of PM2.5 during winter heavy pollution episodes in Beijing from 2013 to 2016, Atmos. Chem. Phys., 18, 247-258, doi: 10.5194/acp-18-247-2018, 2018.